# Point-wise Anomaly Detection via Fold-bifurcation ODE

**Sheo Yon Jhin, Noseong Park**
KAIST
Daejeon, South Korea
sheoyon.jhin@kaist.ac.kr, noseong@kaist.ac.kr

## Abstract

Anomaly detection in time series is essential for applications from industrial monitoring to financial risk management. Recent methods — including forecasting error models, representation learning, augmentation, and weak-label learning — have achieved strong results for specific anomaly types such as sudden point or gradual collective anomalies. While many prior works report window-level metrics that may mask errors, several recent methods evaluate at the point level as well. Our goal is to use a stricter point-wise protocol to make masking effects explicit. We introduce **FOLD** (Point-wise Anomaly Detection via **fold**-bifurcation), a framework that reframes detection as tracking a system's proximity to a critical transition. FOLD extracts stress signals from a forecasting model and integrates them with a fold-bifurcation inspired ODE to produce the risk state, flagging anomalies once it crosses a threshold calibrated on normal data. This requires no anomaly labels and no additional detector training, enabling a parameter-free and efficient detection process. By modeling anomalies as stress accumulation toward a tipping point, FOLD naturally aligns with point-wise detection, providing a unifying and interpretable perspective that complements type-specific methods. Experiments on 40 benchmarks against 34 state-of-the-art baselines show that FOLD achieves competitive or superior performance, with particular strength under strict point-wise evaluation.

## 1 Introduction

Anomaly detection is a fundamental problem with broad impact in domains such as industrial diagnostics, predictive maintenance, and risk management, where the ability to foresee failures is critical for enabling proactive intervention (Chevtchenko et al., 2023; Rodríguez et al., 2023).

Recent methods are largely dominated by two paradigms (Paparrizos et al., 2025): prediction-based approaches, which monitor forecasting or reconstruction errors (Tuli et al., 2022; Xu et al., 2021) and distance-based approaches, which rely on representation learning and embedding similarity (Wang et al., 2025; Deng & Hooi, 2021). However, these paradigms share a common limitation, they primaily capture sudden stress, i.e., sharp deviations at individual timesteps. Prediction-based methods detect instantaneous error spikes, while distance-based models flag sudden embedding shifts. Even when extended over longer horizons, they remain sensitive to momentary fluctuations rather

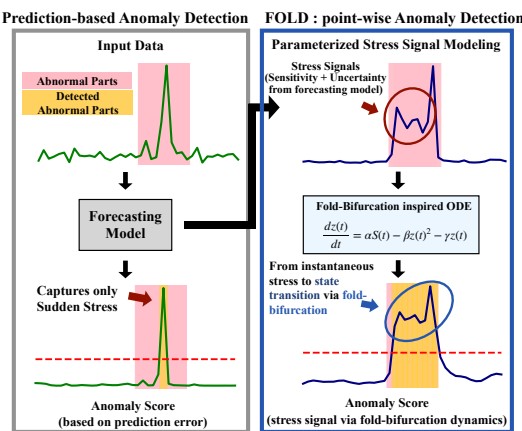

Figure 1: Contrasting conventional prediction-based anomaly detection with our proposed FOLD. While prediction-based models capture only instantaneous error spikes (sudden stress), FOLD extracts sensitivity–uncertainty stress signals and integrates them via fold-bifurcation dynamics, modeling how stress accumulates into state transitions. This enables accurate and interpretable point-wise anomaly detection.

than modeling how stresses accumulate over time. This limitation is often masked by window-level evaluation, where detections are counted correct if they fall anywhere within an anomaly window. Under stricter point-wise anomaly detection, which demands precise localization at every timestep, performance degrades substantially (Wang et al., 2024; Paparrizos et al., 2025; Wang et al., 2025), highlighting the importance of point-wise evaluation as a more faithful and challenging criterion for real-world anomaly detection. Many real-world failures can arise from the accumulation of stress that drives a system toward a critical transition. Importantly, our formulation captures both gradual build-up and short, abrupt spikes within the same dynamical framework. We draw inspiration from fold-bifurcation dynamics, a classical theory in dynamical systems that explains how gradual external pressure can drive a system toward an abrupt transition from normal to failed states. In its canonical form, fold-bifurcation assumes a fixed control parameter $r$, which represents external pressure, and studies how stable and unstable equilibria appear or disappear as $r$ changes. Put simply, the system remains stable until the equilibria collide and vanish, at which point a sudden collapse occurs.

Adapting this principle, we reinterpret the control parameter as a time-varying stress signal $\mathbf{S}(t)$ extracted from a forecasting model. By integrating these stress signals with a fold-bifurcation-inspired ODE, we calculate the risk state $\mathbf{z}(t)$, which captures how small stresses compound into a tipping point, i.e., a critical threshold beyond which the system abruptly shifts from normal to failed behavior. An anomaly is flagged when $\mathbf{z}(t)$ crosses a threshold calibrated solely on normal data.

Building on this perspective, we propose **FOLD** (**Fold**-bifurcation based Anomaly Detection), a framework that reframes detection as modeling stress accumulation, unifying the detection of both sudden deviations and gradual drifts. A distinguishing feature of FOLD is that it can be instantiated directly from an already trained forecasting model without any additional detector training.

We highlight the following contributions of this work:

1. To our knowledge, we introduce **FOLD** the first anomaly detection framework that leverages fold-bifurcation inspired dynamics for point-wise anomaly detection, requiring no anomaly labels and no additional detector training.

2. We provide a principled formulation of anomaly detection as stress-signal driven modeling, where stress signals are integrated through a fold-bifurcation ODE to capture how gradual pressures can accumulate and trigger sudden tipping-point transitions.

3. We conduct extensive experiments on 40 benchmarks against 34 state-of-the-art baselines, demonstrating that FOLD achieves superior performance in both **threshold-dependent** (e.g., Point-wise F1) and **threshold-independent** (e.g., VUS-PR) metrics. This validates the robustness and practical value of our framework under strict point-wise evaluation protocols.

## 2 RELATED WORK

### 2.1 TIME-SERIES ANOMALY DETECTION

Recent advances in time-series anomaly detection can be broadly categorized into two dominant paradigms (Paparrizos et al., 2025): prediction-based methods, which monitor forecasting or reconstruction errors (Tuli et al., 2022; Su et al., 2019; Zhang et al., 2019), and distance-based methods, which rely on representation learning and embedding similarity (Xu et al., 2021; Deng & Hooi, 2021; Wang et al., 2025). While effective under conventional benchmarks, both paradigms share a key limitation: they primarily capture sudden stress, i.e., sharp deviations at individual timesteps. Prediction-based approaches flag error spikes, while distance-based methods detect sudden shifts in representation space. Even when extended to longer horizons, they remain sensitive to momentary fluctuations rather than modeling how stresses accumulate over time.

This limitation is often masked by window-level evaluation, where a detection is considered correct if it falls anywhere within an anomaly window. However, under the stricter point-wise anomaly detection setting, which requires precise localization at each timestep, these methods degrade significantly (Wang et al., 2025; 2024). This explains why many prior approaches report strong results on window-level metrics but fail to generalize under point-wise evaluation (Paparrizos et al., 2025).

Beyond deep neural methods, classical approaches such as Isolation Forest (Liu et al., 2008) and one-class SVMs (Schölkopf et al., 2001) remain widely used in practice, while dynamical change-point detection methods emphasize early-warning indicators in noisy systems. Early-warning studies have also drawn on bifurcation theory, for example by fitting autoregressive models to estimate Jacobian eigenvalues and track critical slowing down in multivariate settings (Williamson & Lenton, 2015). These approaches highlight complementary perspectives, but they are not directly optimized for strict point-wise anomaly detection in complex multivariate data.

Building on these perspectives, FOLD leverages forecasting-derived sensitivity and uncertainty signals as a time-varying stress input to a fold-bifurcation ODE, enabling point-wise anomaly detection through risk-state thresholding. This avoids training an additional detector and instead operates directly on top of a pre-trained forecaster.

## 2.2 Uncertainty Estimation in Neural Time-Series Models

Quantifying predictive uncertainty has become increasingly important in time-series modeling, especially for safety-critical domains where abnormal behavior must be distinguished from normal variability. Practical approaches include Monte Carlo Dropout (Gal & Ghahramani, 2016), which can be interpreted as a Bayesian approximation, and deep ensembles (Lakshminarayanan et al., 2017), which provide scalable estimates of both epistemic and aleatoric uncertainty. These methods not only improve the reliability of point forecasts but also create signals that can be exploited for detecting abnormal or unstable regimes.

Uncertainty has since been directly leveraged in anomaly detection. (Li et al., 2018) used generative models to capture predictive variance in multivariate time series, while (Wiessner et al., 2024) proposed explicit uncertainty-aware detectors, highlighting that high variance is often associated with abnormal or unstable regimes. Beyond machine learning, evidence from complex systems further supports this connection: (Scheffer et al., 2009) showed that variance systematically increases as a system approaches a critical transition, suggesting that rising uncertainty itself can be interpreted as a form of accumulated stress.

In FOLD, we adopt this perspective and treat predictive uncertainty itself as a stress indicator, combining it with sensitivity signals, extending prior variance-based approaches into a unified stress formulation that anticipates anomalies.

## 2.3 Bifurcation Theory and Dynamical Systems Perspective

Bifurcation theory studies how systems can remain stable under gradual change until a critical point (or tipping point) is reached, after which they suddenly shift to a qualitatively different state. A simple example is the fold-bifurcation, where the system has two equilibria (one stable, one unstable) that move closer together as external pressure increases. At a critical threshold, these equilibria collide and vanish, causing the system to abruptly lose stability. Mathematically, this behavior is captured by the canonical equation:

$$\frac{d\mathbf{z}(t)}{dt} = r - \mathbf{z}(t)^2, \tag{1}$$

where $\mathbf{z}(t)$ is the system state and $r$ is a control parameter. Such tipping phenomena have been widely studied in domains such as ecology and climate science (Scheffer, 2009; Lenton et al., 2008). To capture resilience in real systems, this canonical form is often extended with a decay term $\mathbf{z}(t)$:

$$\frac{d\mathbf{z}(t)}{dt} = r - \mathbf{z}(t)^2 - \mathbf{z}(t). \tag{2}$$

While other bifurcation types (e.g., Hopf or pitchfork) describe oscillatory or symmetry-breaking transitions, the fold-bifurcation is particularly suited for anomaly detection because it directly captures the gradual erosion of stability followed by an abrupt collapse. Related early-warning studies have exploited this property by fitting autoregressive models and tracking eigenvalue changes to detect critical slowing down in noisy multivariate systems (Williamson & Lenton, 2015).

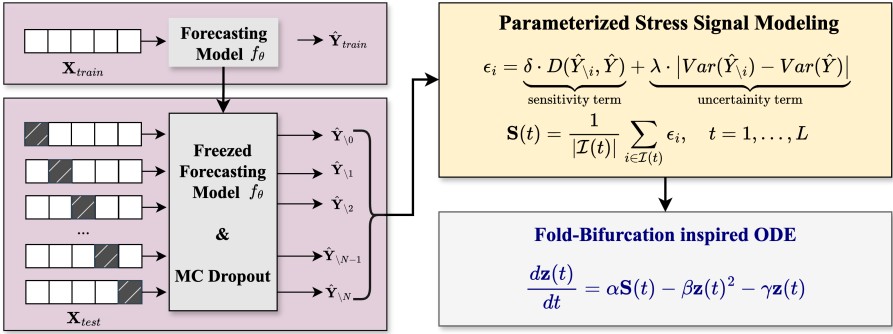

Figure 2: Overall architecture of FOLD.

Building on this adapted form, we reinterpret $r$ as a parameterized time-varying stress signal $\mathbf{S}(t)$ and obtain our fold-bifurcation inspired dynamics:

$$\frac{d\mathbf{z}(t)}{dt} = \alpha\mathbf{S}(t) - \beta\mathbf{z}(t)^2 - \gamma\mathbf{z}(t), \tag{3}$$

where $\alpha, \beta, \gamma$ are fixed coefficients. This formulation preserves the key intuition — stress gradually accumulates, resilience erodes, and a sudden transition occurs — while adapting it for anomaly detection. Unlike early-warning approaches that monitor eigenvalue trends, FOLD instantiates a fold-inspired dynamical mechanism driven by forecasting-derived stress signals to localize anomalies at the point level.

## 3 PROPOSED METHOD

### 3.1 OVERALL WORKFLOW

The workflow of FOLD consists of the following three main steps (cf. Figure 2):

1. A forecasting model $f_\theta$ is first trained on normal sequences using mean squared error (MSE) loss. After training, $f_\theta$ is frozen to serve as a fixed backbone (pink box).

2. For a test sequence, patches are masked and fed into the frozen $f_\theta$ with MC dropout. Sensitivity and uncertainty are combined to form a time-varying stress signal $\mathbf{S}(t)$ (yellow box).

3. The stress signal $\mathbf{S}(t)$ is injected into a fold-bifurcation inspired ODE to evolve the risk trajectory $\mathbf{z}(t)$. Anomalies are flagged once $\mathbf{z}(t)$ leaves its stable basin (gray box).

### 3.2 FORECASTING MODEL AND STRESS SIGNAL SCORING

Let $X = [x_1, \ldots, x_L] \in \mathbb{R}^{L \times d}$ be an input time window of length $L$ with $d$ features. We begin with a forecasting model $f_\theta : \mathbb{R}^{L \times d} \to \mathbb{R}^{H \times d}$, trained solely on normal data using mean squared error (MSE) loss. The model is implemented with dropout layers so that it can later provide Monte Carlo (MC) estimates of predictive uncertainty. After training, $f_\theta$ is frozen for all subsequent steps.

**Perturbed prediction** $\hat{Y}_{\backslash i}$. The input sequence $X = [x_1, \ldots, x_L]$ is divided into $N$ patches $\{P_i\}_{i=1}^N$. For each patch $P_i$, we mask out (i.e., replace with a mask token or zero) the corresponding segment and feed the modified sequence $X_{\backslash P_i}$ into the frozen forecasting model $f_\theta$, $\hat{Y}_{\backslash i} = f_\theta(X_{\backslash P_i})$ (cf. Figure 2).

**Sensitivity term.** To measure how much the forecast depends on a given patch, we compute a distance between the baseline prediction $\hat{Y} = f_\theta(X)$ and the perturbed prediction $\hat{Y}_{\backslash i}$: $D(\hat{Y}_{\backslash i}, \hat{Y})$, where $D(\cdot, \cdot)$ is a distance metric (e.g., MSE, MAE, and cosine similarity). The rationale is to capture how strongly each local patch influences future predictions — patches whose removal causes large deviations in $\hat{Y}$ are assigned higher sensitivity. We provide an ablation in Appendix E.1 to show the effect of different distance metrics.

**Uncertainty term.** To caputre how unstable the model becomes under local perturbations, we estimate predictive variance via Monte Carlo dropout. For each input, we run $T_{MC}$ stochastic forward passes: $\left|Var(\hat{Y}_{\setminus i}) - Var(\hat{Y})\right|$. The variance difference measures the additional epistemic and aleatoric uncertainity introduced when the patch is masked. The use of MC dropout is motivated by Bayesian approximations in neural networks (Lakshminarayanan et al., 2017), as discussed in Section 2.2.

**Stress Signal.** Combining the two components yields the patch-level stress score:

$$\epsilon_i = \underbrace{\delta \cdot D(\hat{Y}_{\setminus i}, \hat{Y})}_{\text{sensitivity term}} + \underbrace{\lambda \cdot \left|Var(\hat{Y}_{\setminus i}) - Var(\hat{Y})\right|}_{\text{uncertainty term}}, \quad \epsilon_i \in \mathbb{R}^{L \times d}, \tag{4}$$

where $\delta, \lambda > 0$ control the relative importance of sensitivity and uncertainty. Ablation studies in Section 5 confirm that both terms are complementary: omitting either degrades performance.

**Sequence-level stress signal.** Finally, we aggregate patch-level scores into a time-varying signal aligned with the original input:

$$\mathbf{S}(t) = \frac{1}{|\mathcal{I}(t)|} \sum_{i \in \mathcal{I}(t)} \epsilon_i, \quad \mathbf{S}(t) \in \mathbb{R}^d, \; t = 1, \ldots, L, \tag{5}$$

where $\mathcal{I}(t)$ is the set of patches covering index $t$. This produce a stress signal $\mathbf{S} \in \mathbb{R}^{L \times d}$ with the same temporal and feature dimensions as the input sequence, serving as the data-driven analogue of external pressure in the fold-bifurcation dynamics.

### 3.3 FOLD-BIFURCATION INSPIRED ODE MODELING

Real-world systems often exhibit tipping-point behavior: they appear stable while absorbing small stresses, but once a critical threshold is crossed, they abruptly transition to a failed state. This phenomenon is mathematically captured by fold-bifurcation, where the canonical equation

$$\frac{d\mathbf{z}(t)}{dt} = r - \mathbf{z}(t)^2 \tag{6}$$

shows how gradual changes in a control parameter $r$ can eliminate equilibria and precipitate sudden collapse. Among canonical bifurcations, we adopt the fold (saddle–node) because our targets are abrupt losses of stability under monotone external pressure rather than oscillatory onsets (Hopf) or symmetry breaking (pitchfork). The fold directly captures equilibrium annihilation under gradually increasing pressure, which matches point-wise anomaly onset in practice.

FOLD builds directly on this intuition. Consistent with the stress-centric view in Section 2.3, we reinterpret the fixed parameter $r$ as the time-varying stress signal $\mathbf{S}(t)$ derived from the forecasting model. Conceptually, just as increasing $r$ leads to a collapse in the canonical form, a rising stress signal $\mathbf{S}(t)$ acts as a dynamic external force that drives the system state toward instability. This yields our fold-bifurcation inspired dynamics of the risk state $\mathbf{z}(t) \in \mathbb{R}^d$:

$$\frac{d\mathbf{z}(t)}{dt} = \alpha \mathbf{S}(t) - \beta \mathbf{z}(t)^2 - \gamma \mathbf{z}(t), \tag{7}$$

where the dynamics are computed feature-wise over $d$ dimensions, resulting in a full trajectory $\mathbf{z} \in \mathbb{R}^{L \times d}$ across the window. To obtain a unified anomaly indicator for detection, we aggregate these feature-wise trajectories into a system-level risk score $\mathbf{z}_{sys} \in \mathbb{R}^L$.

**Interpretation of coefficients.** In this Eq. 7, $\mathbf{S}(t)$ replaces the constant control parameter $r$ and acts as a data-driven analogue of external pressure, injected feature-wise from the forecasting model. The coefficient $\alpha > 0$ scales stress injection, $\gamma > 0$ provides resilience by pulling the system back toward stability when stress subsides, and $\beta > 0$ induces nonlinear escalation, where accumulated risk amplifies disproportionately near a tipping point. Together, these terms instantiate a fold-bifurcation inspired mechanism of stress accumulation leading to critical transitions.

**Numerical solution.** Since Eq. 7 has no closed-form solution, we compute $\mathbf{z}(t)$ using standard adaptive ODE solvers (e.g., Runge–Kutta). These solvers provide stable integration under time-varying stress signals, ensuring that anomaly detection reflects the dynamics of the model rather than artifacts of discretization.

**Free anomaly detection.** The coefficients $(\alpha, \beta, \gamma)$ act as fixed hyperparameters, selected based on data statistics rather than learned from anomaly labels. Anomalies are flagged once $\mathbf{z}(t)$ exits its stable basin under the influence of $\mathbf{S}(t)$. This realizes the principle of parameter-free detection: given a forecasting model, anomalies are identified directly through principled dynamical modeling without any detector-specific optimization. Furthermore, when integrated with pre-trained foundation models (e.g., Chronos) as demonstrated in our experiments, this framework extends to a fully training-free (zero-shot) anomaly detection pipeline, eliminating the need for backbone training as well.

### 3.4 ANOMALY CRITERION

To derive a stability boundary, we simulate Eq. 7 on normal training data. Because the forecasting model $f_\theta$ involves stochasticity (e.g., dropout), we generate multiple risk trajectories $\mathbf{z}(t)$ (or $\mathbf{z}_{sys}(t)$ for multivariate settings) by varying the random seed. For each normal risk states $\mathbf{z}(t)$, we then record its maximum value, yielding

$$M_{\text{train}} = \{\max_t \mathbf{z}(t) \mid X \in \mathcal{D}_{\text{train}}^{\text{normal}}\}. \tag{8}$$

Following the fold-bifurcation intuition, an anomaly should only be declared once the risk state exceeds the typical maxima observed in normal regimes. To capture this, we define the threshold as a high quantile of the normal maxima with a small multiplicative margin:

$$Z_{\text{thr}} = (1 + \rho) \cdot \text{Quantile}_p(M_{\text{train}}), \quad p \approx 0.95\text{--}0.99, \ \rho \approx 0.05. \tag{9}$$

The quantile term ensures robustness against rare fluctuations, while the margin $\rho$ reflects the bifurcation principle that the system must not merely touch but clearly surpass the stability boundary before being considered anomalous. Conceptually, we interpret this calibrated threshold $Z_{thr}$ as the empirical boundary of the stable basin. Therefore, the event $\mathbf{z}(t) > Z_{thr}$ (or $\mathbf{z}_{sys}(t) > Z_{thr}$ for multivariate settings) serves as the operational criterion for determining that the system has overcome its resilience and crossed the tipping point.

**Point-wise decision rule.** For a test sequence $X$ with trajectory $\mathbf{z}_{sys}(t)$, we then produce a point-wise anomaly mask:

$$\hat{y}(t) = \mathbb{1}\{\mathbf{z}_{sys}(t) > Z_{\text{thr}}\}, \quad t = 1, \dots, L. \tag{10}$$

All metrics (precision, recall, and F1) are computed at the timestep level. This rule provides a statistically grounded and label-free boundary for point-wise anomaly detection, as illustrated in Figure 3. To further assess robustness of the calibrated threshold $Z_{\text{thr}}$, we conduct an ablation study under $\varepsilon$-contamination, with results reported in Section 5 and Appendix E.4. (Note: While this threshold enables binary decisions, we primarily evaluate performance using the threshold-independent VUS-PR metric in Section 4 to demonstrate the model's intrinsic robustness.)

## 4 EXPERIMENTS

We conduct a comprehensive evaluation on the **TSB-AD benchmark** (Liu & Paparrizos, 2024), a widely recognized leaderboard comprising 40 curated datasets (1,070 time series in total). To ensure a rigorous and unbiased assessment, we adopt VUS-PR (Volume Under the Surface of Precision-Recall curve) (Paparrizos et al., 2022) as our primary evaluation metric. Unlike standard F1-scores, VUS-PR is threshold-independent, effectively eliminating potential biases arising from specific threshold selections. For completeness, we also provide threshold-dependent results (e.g., Point-wise F1). This stricter protocol replaces window-level scoring common in prior work. See Appendix B for dataset, hyperparameter, and evaluation details.

Table 1: **VUS-PR score (Higher is better) averaged over all time series for each dataset** for univariate anomaly detection. The best VUS-PR are shown in **bold**, and the second-best in *italic*. Implementation details about **FOLD**(Chronos) are in Appendix B.5.

| Method | CATSv2 | Daphnet | Exathlon | IOPS | LTDB | MGAB | MITDB | MSL | NAB | NEK | OPPORTUNITY | Power | SED | SMAP | SMD | SVDB | SWaT | Stock | TAO | TODS | UCR | WSD | YAHOO | Avg. RANK |
|---|---|---|---|---|---|---|---|---|---|---|---|---|---|---|---|---|---|---|---|---|---|---|---|---|
| **FOLD (DLinear)** | 0.42 | 0.57 | 0.95 | 0.63 | 0.59 | 0.05 | 0.22 | 0.70 | 0.56 | 0.92 | 0.94 | 0.18 | 0.29 | 0.82 | 0.84 | 0.31 | 0.58 | 0.99 | 1.00 | 0.79 | 0.32 | 0.65 | 0.85 | 3.86 |
| **FOLD (Chronos)** | 0.40 | 0.55 | 0.97 | 0.61 | 0.81 | 0.14 | 0.31 | 0.64 | 0.62 | 0.94 | 0.97 | 0.14 | 0.27 | 0.84 | 0.87 | 0.37 | 0.69 | 1.00 | 1.00 | 0.84 | 0.38 | 0.58 | 0.83 | 2.95 |
| TSPulse (FT) | 0.40 | 0.54 | 0.87 | 0.42 | 0.52 | 0.01 | 0.22 | 0.66 | 0.54 | 0.50 | 0.58 | 0.06 | 0.08 | 0.07 | 0.74 | 0.80 | 0.56 | 0.10 | 0.98 | 0.68 | 0.28 | 0.43 | 0.83 | 8.65 |
| TSPulse (ZS) | 0.35 | 0.55 | 0.83 | 0.42 | 0.41 | 0.00 | 0.10 | 0.64 | 0.50 | 0.58 | 0.06 | 0.07 | 0.06 | 0.71 | 0.78 | 0.36 | 0.10 | 0.98 | 1.00 | 0.68 | 0.18 | 0.42 | 0.83 | 11.30 |
| Sub-PCA | 0.26 | 0.42 | 0.93 | 0.23 | 0.56 | 0.01 | 0.36 | 0.51 | 0.44 | 0.91 | 0.91 | 0.08 | 0.03 | 0.52 | 0.45 | 0.52 | 0.39 | 0.84 | 0.93 | 0.54 | 0.12 | 0.09 | 0.14 | 13.39 |
| KShapeAD | 0.25 | 0.04 | 0.33 | 0.09 | 0.83 | 0.02 | 0.69 | 0.55 | 0.37 | 0.24 | 0.33 | 0.19 | 0.89 | 0.58 | 0.13 | 0.82 | 0.43 | 0.75 | 0.91 | 0.75 | 0.38 | 0.10 | 0.55 | 13.69 |
| POLY | 0.23 | 0.51 | 0.74 | 0.31 | 0.51 | 0.01 | 0.34 | 0.54 | 0.48 | 0.61 | 0.10 | 0.09 | 0.04 | 0.64 | 0.61 | 0.44 | 0.10 | 0.82 | 0.92 | 0.57 | 0.13 | 0.41 | 0.25 | 13.60 |
| Series2Graph | 0.21 | 0.19 | 0.60 | 0.22 | 0.79 | 0.00 | 0.61 | 0.25 | 0.44 | 0.67 | 0.11 | 0.07 | 0.15 | 0.55 | 0.46 | 0.55 | 0.22 | 0.79 | 0.91 | 0.73 | 0.25 | 0.27 | 0.28 | 14.26 |
| MOMENT (FT) | 0.38 | 0.51 | 0.83 | 0.38 | 0.45 | 0.00 | 0.13 | 0.53 | 0.39 | 0.73 | 0.07 | 0.07 | 0.04 | 0.63 | 0.75 | 0.23 | 0.08 | 0.81 | 0.94 | 0.58 | 0.09 | 0.50 | 0.25 | 14.69 |
| MOMENT (ZS) | 0.30 | 0.52 | 0.81 | 0.37 | 0.44 | 0.00 | 0.14 | 0.53 | 0.39 | 0.73 | 0.07 | 0.08 | 0.04 | 0.62 | 0.74 | 0.27 | 0.07 | 0.81 | 0.94 | 0.58 | 0.07 | 0.49 | 0.23 | 14.91 |
| KMeansAD | 0.23 | 0.04 | 0.41 | 0.06 | 0.49 | 0.01 | 0.27 | 0.48 | 0.33 | 0.20 | 0.30 | 0.39 | 0.87 | 0.63 | 0.18 | 0.44 | 0.10 | 0.76 | 0.92 | 0.65 | 0.38 | 0.10 | 0.56 | 16.00 |
| USAD | 0.40 | 0.12 | 0.89 | 0.13 | 0.55 | 0.00 | 0.18 | 0.27 | 0.28 | 0.73 | 0.67 | 0.06 | 0.03 | 0.27 | 0.66 | 0.43 | 0.37 | 0.75 | 0.93 | 0.52 | 0.08 | 0.04 | 0.10 | 18.65 |
| Sub-KNN | 0.29 | 0.04 | 0.47 | 0.10 | 0.58 | 0.24 | 0.36 | 0.33 | 0.29 | 0.23 | 0.30 | 0.21 | 0.87 | 0.51 | 0.14 | 0.56 | 0.10 | 0.75 | 0.92 | 0.65 | 0.37 | 0.10 | 0.31 | 15.82 |
| MatrixProfile | 0.36 | 0.04 | 0.56 | 0.10 | 0.58 | 0.29 | 0.39 | 0.48 | 0.32 | 0.13 | 0.25 | 0.15 | 0.72 | 0.47 | 0.13 | 0.36 | 0.11 | 0.72 | 0.92 | 0.76 | 0.34 | 0.02 | 0.43 | 16.08 |
| SAND | 0.27 | 0.04 | 0.25 | 0.06 | 0.79 | 0.01 | 0.67 | 0.30 | 0.38 | 0.32 | 0.18 | 0.16 | 0.75 | 0.56 | 0.11 | 0.72 | 0.21 | 0.74 | 0.91 | 0.70 | 0.34 | 0.08 | 0.41 | 16.60 |
| CNN | 0.32 | 0.40 | 0.61 | 0.26 | 0.42 | 0.01 | 0.15 | 0.33 | 0.19 | 0.73 | 0.40 | 0.06 | 0.06 | 0.34 | 0.55 | 0.21 | 0.68 | 0.92 | 1.00 | 0.68 | 0.54 | 0.05 | 0.24 | 13.86 |
| LSTMAD | 0.33 | 0.13 | 0.73 | 0.20 | 0.36 | 0.03 | 0.12 | 0.32 | 0.18 | 0.73 | 0.58 | 0.07 | 0.06 | 0.26 | 0.49 | 0.13 | 0.67 | 0.85 | 1.00 | 0.47 | 0.02 | 0.13 | 0.45 | 16.47 |
| SR | 0.28 | 0.20 | 0.73 | 0.24 | 0.29 | 0.01 | 0.07 | 0.22 | 0.20 | 0.50 | 0.33 | 0.10 | 0.07 | 0.29 | 0.36 | 0.08 | 0.35 | 1.00 | 1.00 | 0.64 | 0.07 | 0.22 | 0.61 | 16.17 |
| TimesFM | 0.25 | 0.36 | 0.53 | 0.20 | 0.27 | 0.00 | 0.06 | 0.32 | 0.18 | 0.35 | 0.05 | 0.08 | 0.05 | 0.30 | 0.40 | 0.06 | 0.22 | 0.99 | 0.99 | 0.75 | 0.07 | 0.21 | 0.81 | 19.21 |
| IForest | 0.08 | 0.36 | 0.67 | 0.28 | 0.34 | 0.00 | 0.10 | 0.29 | 0.22 | 0.59 | 0.43 | 0.06 | 0.36 | 0.25 | 0.34 | 0.09 | 0.50 | 0.99 | 0.99 | 0.52 | 0.02 | 0.14 | 0.44 | 17.21 |
| OmniAnomaly | 0.12 | 0.16 | 0.83 | 0.20 | 0.32 | 0.00 | 0.10 | 0.25 | 0.19 | 0.85 | 0.60 | 0.07 | 0.06 | 0.15 | 0.36 | 0.09 | 0.44 | 0.82 | 0.98 | 0.44 | 0.03 | 0.14 | 0.19 | 19.86 |
| Lag-Llama | 0.21 | 0.39 | 0.53 | 0.22 | 0.29 | 0.00 | 0.08 | 0.31 | 0.18 | 0.38 | 0.05 | 0.08 | 0.07 | 0.28 | 0.36 | 0.08 | 0.09 | 0.97 | 0.99 | 0.61 | 0.02 | 0.22 | 0.68 | 20.21 |
| Chronos | 0.10 | 0.31 | 0.45 | 0.18 | 0.26 | 0.00 | 0.06 | 0.18 | 0.18 | 0.34 | 0.06 | 0.08 | 0.06 | 0.19 | 0.32 | 0.06 | 0.14 | 0.99 | 1.00 | 0.70 | 0.07 | 0.18 | 0.80 | 21.08 |
| TimesNet | 0.10 | 0.39 | 0.53 | 0.22 | 0.29 | 0.00 | 0.08 | 0.31 | 0.20 | 0.37 | 0.05 | 0.08 | 0.05 | 0.38 | 0.54 | 0.09 | 0.11 | 0.79 | 0.91 | 0.59 | 0.02 | 0.27 | 0.29 | 21.00 |
| AutoEncoder | 0.18 | 0.09 | 0.36 | 0.25 | 0.69 | 0.01 | 0.07 | 0.27 | 0.32 | 0.51 | 0.12 | 0.09 | 0.41 | 0.49 | 0.14 | 0.32 | 0.38 | 0.72 | 0.93 | 0.65 | 0.09 | 0.14 | 0.29 | 17.52 |
| TranAD | 0.08 | 0.13 | 0.72 | 0.18 | 0.31 | 0.00 | 0.09 | 0.18 | 0.18 | 0.72 | 0.58 | 0.07 | 0.05 | 0.13 | 0.16 | 0.09 | 0.46 | 0.79 | 0.94 | 0.45 | 0.02 | 0.11 | 0.28 | 22.17 |
| FITS | 0.17 | 0.43 | 0.55 | 0.17 | 0.34 | 0.00 | 0.09 | 0.36 | 0.24 | 0.49 | 0.07 | 0.07 | 0.05 | 0.42 | 0.54 | 0.10 | 0.10 | 0.76 | 0.91 | 0.58 | 0.02 | 0.14 | 0.18 | 21.20 |
| Sub-LOF | 0.31 | 0.04 | 0.25 | 0.11 | 0.34 | 0.44 | 0.26 | 0.35 | 0.32 | 0.25 | 0.12 | 0.14 | 0.22 | 0.40 | 0.04 | 0.18 | 0.11 | 0.76 | 0.92 | 0.53 | 0.29 | 0.03 | 0.27 | 19.86 |
| OFA | 0.16 | 0.36 | 0.55 | 0.20 | 0.30 | 0.00 | 0.07 | 0.29 | 0.21 | 0.37 | 0.05 | 0.08 | 0.06 | 0.33 | 0.45 | 0.07 | 0.11 | 0.76 | 0.91 | 0.54 | 0.02 | 0.16 | 0.24 | 22.52 |
| Sub-MCD | 0.37 | 0.04 | 0.23 | 0.13 | 0.24 | 0.01 | 0.11 | 0.16 | 0.19 | 0.11 | 0.32 | 0.30 | 0.12 | 0.30 | 0.08 | 0.07 | 0.09 | 0.75 | 0.90 | 0.64 | 0.26 | 0.15 | 0.28 | 23.08 |
| Sub-HBOS | 0.04 | 0.05 | 0.45 | 0.05 | 0.69 | 0.00 | 0.17 | 0.25 | 0.30 | 0.23 | 0.08 | 0.12 | 0.88 | 0.55 | 0.10 | 0.24 | 0.12 | 0.70 | 0.93 | 0.64 | 0.14 | 0.01 | 0.06 | 21.91 |
| Sub-OCSVM | 0.26 | 0.06 | 0.29 | 0.07 | 0.33 | 0.01 | 0.14 | 0.28 | 0.26 | 0.26 | 0.11 | 0.11 | 0.16 | 0.06 | 0.11 | 0.16 | 0.20 | 0.73 | 0.92 | 0.65 | 0.18 | 0.03 | 0.23 | 22.00 |
| Sub-IForest | 0.05 | 0.07 | 0.49 | 0.04 | 0.66 | 0.00 | 0.24 | 0.36 | 0.30 | 0.22 | 0.07 | 0.12 | 0.79 | 0.47 | 0.09 | 0.27 | 0.13 | 0.69 | 0.90 | 0.66 | 0.10 | 0.01 | 0.06 | 22.04 |
| Donut | 0.08 | 0.06 | 0.45 | 0.10 | 0.31 | 0.00 | 0.10 | 0.20 | 0.18 | 0.47 | 0.18 | 0.09 | 0.14 | 0.31 | 0.29 | 0.08 | 0.47 | 0.78 | 0.91 | 0.48 | 0.01 | 0.06 | 0.12 | 23.91 |
| LOF | 0.06 | 0.13 | 0.20 | 0.12 | 0.26 | 0.00 | 0.06 | 0.15 | 0.17 | 0.38 | 0.14 | 0.09 | 0.11 | 0.15 | 0.13 | 0.05 | 0.12 | 0.75 | 0.91 | 0.49 | 0.02 | 0.09 | 0.37 | 26.04 |
| AnomalyTransformer | 0.05 | 0.07 | 0.13 | 0.06 | 0.27 | 0.00 | 0.09 | 0.14 | 0.14 | 0.23 | 0.07 | 0.09 | 0.09 | 0.09 | 0.18 | 0.07 | 0.10 | 0.75 | 0.90 | 0.46 | 0.01 | 0.02 | 0.07 | 29.26 |

Table 2: **VUS-PR score (Higher is better) averaged over all time series for each dataset** for multivariate anomaly detection.

| Method | CATSv2 | CreditCard | Daphnet | Exathlon | GECCO | GHL | Genesis | LTDB | MITDB | MSL | OPPORTUNITY | PSM | SMAP | SMD | SVDB | SWaT | TAO | Avg. RANK |
|---|---|---|---|---|---|---|---|---|---|---|---|---|---|---|---|---|---|---|
| **FOLD (ours)** | 0.23 | 0.19 | 0.39 | 0.93 | 0.08 | 0.07 | 0.38 | 0.37 | 0.09 | 0.50 | 0.82 | 0.19 | 0.45 | 0.46 | 0.39 | 0.50 | 0.97 | 3.11 |
| TSPulse (FT) | 0.07 | 0.00 | 0.35 | 0.91 | 0.18 | 0.01 | 0.02 | 0.57 | 0.14 | 0.21 | 0.07 | 0.14 | 0.32 | 0.36 | 0.47 | 0.14 | 0.93 | 9.88 |
| TSPulse (ZS) | 0.05 | 0.00 | 0.35 | 0.89 | 0.17 | 0.01 | 0.01 | 0.36 | 0.07 | 0.20 | 0.07 | 0.14 | 0.30 | 0.35 | 0.38 | 0.13 | 0.93 | 12.17 |
| CNN | 0.08 | 0.02 | 0.21 | 0.68 | 0.03 | 0.02 | 0.10 | 0.33 | 0.14 | 0.35 | 0.16 | 0.22 | 0.19 | 0.35 | 0.19 | 0.41 | 1.00 | 7.52 |
| OmniAnomaly | 0.04 | 0.02 | 0.34 | 0.84 | 0.02 | 0.07 | 0.00 | 0.44 | 0.11 | 0.22 | 0.18 | 0.16 | 0.12 | 0.17 | 0.35 | 0.15 | 0.81 | 11.17 |
| PCA | 0.12 | 0.10 | 0.13 | 0.95 | 0.20 | 0.01 | 0.02 | 0.24 | 0.07 | 0.15 | 0.30 | 0.16 | 0.09 | 0.36 | 0.11 | 0.45 | 1.00 | 9.41 |
| LSTMAD | 0.04 | 0.02 | 0.31 | 0.82 | 0.02 | 0.06 | 0.04 | 0.30 | 0.09 | 0.22 | 0.17 | 0.24 | 0.16 | 0.33 | 0.15 | 0.16 | 0.99 | 9.41 |
| USAD | 0.04 | 0.02 | 0.34 | 0.84 | 0.02 | 0.06 | 0.00 | 0.41 | 0.12 | 0.23 | 0.18 | 0.19 | 0.11 | 0.16 | 0.32 | 0.15 | 0.81 | 11 |
| AutoEncoder | 0.06 | 0.03 | 0.13 | 0.91 | 0.05 | 0.05 | 0.01 | 0.21 | 0.04 | 0.22 | 0.14 | 0.28 | 0.13 | 0.30 | 0.06 | 0.58 | 1.00 | 10.17 |
| KMeansAD | 0.12 | 0.02 | 0.10 | 0.37 | 0.06 | 0.03 | 0.89 | 0.41 | 0.06 | 0.21 | 0.06 | 0.38 | 0.36 | 0.16 | 0.20 | 0.16 | 0.86 | 8.58 |
| CBLOF | 0.06 | 0.03 | 0.10 | 0.86 | 0.03 | 0.02 | 0.02 | 0.20 | 0.04 | 0.21 | 0.14 | 0.19 | 0.14 | 0.22 | 0.07 | 0.29 | 1.00 | 13.11 |
| MCD | 0.13 | 0.06 | 0.14 | 0.80 | 0.03 | 0.01 | 0.06 | 0.21 | 0.04 | 0.23 | 0.17 | 0.26 | 0.10 | 0.26 | 0.07 | 0.54 | 1.00 | 10.11 |
| OCSVM | 0.08 | 0.02 | 0.06 | 0.83 | 0.04 | 0.04 | 0.08 | 0.20 | 0.04 | 0.22 | 0.12 | 0.19 | 0.12 | 0.28 | 0.06 | 0.44 | 0.81 | 12.64 |
| Donut | 0.07 | 0.02 | 0.17 | 0.66 | 0.03 | 0.05 | 0.18 | 0.26 | 0.12 | 0.30 | 0.15 | 0.20 | 0.18 | 0.19 | 0.11 | 0.44 | 0.75 | 10.23 |
| RobustPCA | 0.04 | 0.02 | 0.06 | 0.77 | 0.02 | 0.03 | 0.00 | 0.23 | 0.04 | 0.22 | 0.13 | 0.12 | 0.07 | 0.10 | 0.08 | 0.12 | 1.00 | 17.17 |
| FITS | 0.13 | 0.02 | 0.33 | 0.63 | 0.03 | 0.01 | 0.01 | 0.23 | 0.05 | 0.17 | 0.05 | 0.13 | 0.08 | 0.17 | 0.10 | 0.15 | 0.78 | 15.41 |
| OFA | 0.13 | 0.02 | 0.31 | 0.58 | 0.04 | 0.01 | 0.22 | 0.29 | 0.06 | 0.14 | 0.05 | 0.17 | 0.08 | 0.17 | 0.12 | 0.12 | 0.78 | 14.52 |
| EIF | 0.06 | 0.02 | 0.15 | 0.41 | 0.04 | 0.02 | 0.06 | 0.19 | 0.04 | 0.18 | 0.10 | 0.18 | 0.13 | 0.32 | 0.07 | 0.32 | 0.89 | 14 |
| COPOD | 0.05 | 0.05 | 0.21 | 0.40 | 0.04 | 0.03 | 0.08 | 0.21 | 0.04 | 0.21 | 0.17 | 0.20 | 0.10 | 0.19 | 0.07 | 0.31 | 0.99 | 12.52 |
| IForest | 0.05 | 0.03 | 0.13 | 0.35 | 0.04 | 0.05 | 0.08 | 0.21 | 0.04 | 0.21 | 0.18 | 0.19 | 0.09 | 0.26 | 0.07 | 0.39 | 0.93 | 53.52 |
| HBOS | 0.05 | 0.04 | 0.15 | 0.32 | 0.04 | 0.04 | 0.08 | 0.21 | 0.04 | 0.23 | 0.17 | 0.17 | 0.09 | 0.25 | 0.07 | 0.30 | 0.83 | 12.88 |
| TimesNet | 0.07 | 0.02 | 0.27 | 0.42 | 0.03 | 0.01 | 0.04 | 0.27 | 0.07 | 0.17 | 0.06 | 0.14 | 0.09 | 0.14 | 0.11 | 0.14 | 0.79 | 15.88 |
| KNN | 0.07 | 0.02 | 0.25 | 0.33 | 0.11 | 0.01 | 0.04 | 0.19 | 0.04 | 0.18 | 0.06 | 0.12 | 0.12 | 0.30 | 0.06 | 0.11 | 0.78 | 16.82 |
| TranAD | 0.04 | 0.02 | 0.31 | 0.10 | 0.02 | 0.06 | 0.04 | 0.26 | 0.07 | 0.24 | 0.16 | 0.23 | 0.09 | 0.30 | 0.12 | 0.15 | 0.81 | 12.05 |
| LOF | 0.05 | 0.02 | 0.11 | 0.16 | 0.13 | 0.01 | 0.08 | 0.19 | 0.04 | 0.14 | 0.10 | 0.15 | 0.09 | 0.16 | 0.06 | 0.15 | 0.79 | 17.52 |
| AnomalyTransformer | 0.03 | 0.02 | 0.07 | 0.10 | 0.02 | 0.03 | 0.01 | 0.21 | 0.05 | 0.12 | 0.07 | 0.21 | 0.06 | 0.07 | 0.08 | 0.18 | 0.77 | 19 |

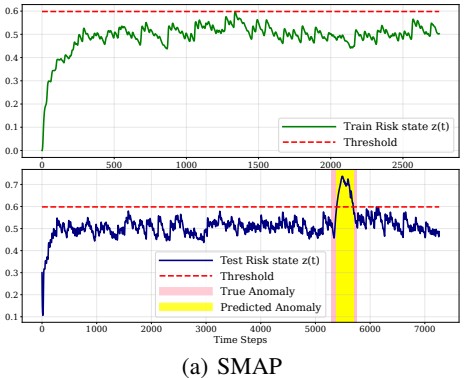 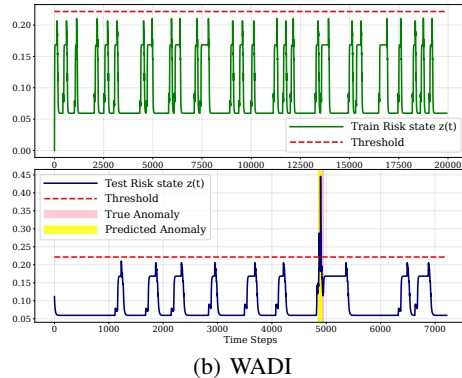

(a) SMAP (b) WADI

Figure 3: Illustration of threshold calibration on three datasets. For each dataset, the top panel shows the trajectories of risk states $\mathbf{z}_{sys}(t)$ on normal training data, with the red dashed line indicating the chosen high-percentile threshold $Z_{\text{thr}}$. The bottom panel shows the risk state trajectories on test data, where anomalies are flagged once $\mathbf{z}_{sys}(t)$ exceeds this threshold.

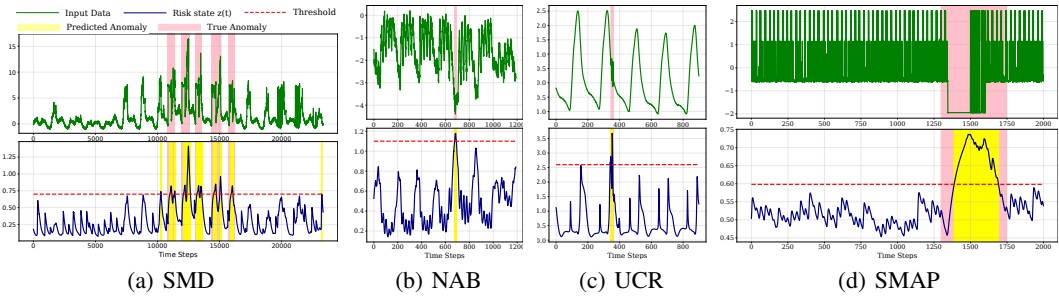

(a) SMD (b) NAB (c) UCR (d) SMAP

Figure 4: Visualization of anomaly detection results on 4 benchmark datasets. Additional visualizations are provided in Appendix G.

## 4.1 EXPERIMENTAL RESULTS

We conducted a comprehensive evaluation on the full TSB-AD benchmark suite (40 datasets) using the threshold-independent VUS-PR metric. Additional experimental results are provided in Appendix C.

**Univariate Performance and Foundation Model Synergy.** FOLD achieves the best average rank of 3.86 across 23 datasets, significantly outperforming the runner-up TSPulse (Avg Rank 8.65). This validates our mechanism's dual capability: high-magnitude stress $\mathbf{S}(t)$ instantly overcomes damping $\gamma$ to detect abrupt spikes (e.g., NAB), while the nonlinear term $-\beta z^2$ integrates gradual accumulation (e.g., SMAP) to trigger tipping points. Furthermore, integrating FOLD with Chronos (Ansari et al., 2024) in a zero-shot setting yields an even superior rank of 2.95. This demonstrates that our framework acts as a universal detection mechanism, effectively transforming probabilistic outputs — whether from lightweight backbones or foundation models — into a robust external force that drives dynamical state transitions without additional detector training.

**Multivariate Performance.** In the multivariate track, FOLD maintains the top position with an average rank of 3.11, surpassing deep learning baselines such as CNN (Avg Rank 7.52). While deep learning methods often struggle with noise amplification in high-dimensional spaces, FOLD leverages the forecasting backbone to encode inter-variable dependencies into the stress signal. By aggregating these feature-wise risk dynamics into a system-level score, FOLD effectively filters out isolated channel noise while amplifying synchronized stress events. This results in superior scalability and stability on complex, highly correlated systems like SWaT and OPPORTUNITY, where maintaining low false positives is crucial.

**Overall Robustness.** The analysis highlights a key distinction: while statistical baselines (e.g., Sub-PCA) rely on rigid linearity assumptions — performing well only on simple stationary datasets — FOLD's dynamical formulation naturally adapts to nonlinear transitions across varying domains. Furthermore, compared to foundation model-based methods like MOMENT or TSPulse, FOLD achieves higher consistency without the computational burden of extensive pre-training or fine-tuning. This suggests that a principled dynamical mechanism can be more effective and efficient than purely data-driven scale in capturing fundamental anomaly characteristics.

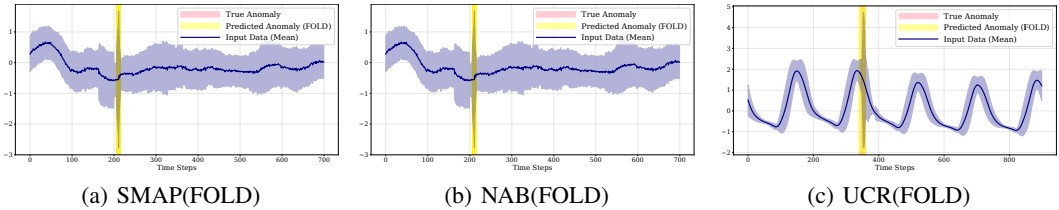

(a) SMAP(FOLD)  (b) NAB(FOLD)  (c) UCR(FOLD)

Figure 5: Visualization of uncertainty assessments to demonstrate reliability and confidence. The blue line represents the input data, and the shaded region indicates the uncertainty band (derived from MC dropout). Note that the uncertainty band remains narrow during normal states (indicating high confidence) but significantly widens at the onset of anomalies, triggering the risk accumulation mechanism. This visually confirms that FOLD's detections are driven by model-intrinsic risk assessment rather than random fluctuations.

**Visualizations and Uncertainty.** Figure 4 demonstrates FOLD's detection performance across diverse failure modes. The model accurately tracks gradual stress accumulation (e.g., SMAP) as well as abrupt spikes (e.g., NAB, UCR), effectively triggering alerts when the risk state $\mathbf{z}(t)$ (or $\mathbf{z}_{sys}(t)$ for multivariate setting) crosses the threshold. To further validate reliability, Figure 5 explicitly visualizes the predictive uncertainty bands (shaded regions derived from MC dropout). As shown, these bands remain narrow during normal states (indicating high confidence) but significantly widen at the onset of anomalies. This visual evidence confirms that FOLD's detections are driven by a model-intrinsic increase in risk and uncertainty, rather than random fluctuations, ensuring robust decision-making.

## 5 ABLATION AND SENSITIVITY STUDIES

**Impact of stress signal components.** As motivated in Section 3.2, we design the stress signal by combining two terms: sensitivity, which reflects how fragile forecasts are to local perturbations, and uncertainty, which captures how unstable the model becomes under such perturbations. Both have been independently validated in prior work as meaningful indicators of anomalous behavior. Table 4 presents an ablation study confirming their complementary roles. Removing the uncertainty term leads to excessive false positives, as the model reacts strongly to transient fluctuations. Conversely, removing the sensitivity term causes under-detection, since uncertainty alone cannot capture sharp deviations. Only when combined do the two signals yield a balanced and robust quantification of stress, enabling the full FOLD model to achieve the highest F1-scores across benchmarks. This result substantiates our formulation: anomalies are best captured when both local fragility and systemic instability are jointly considered as drivers of stress accumulation.

**Calibration robustness.** Since our anomaly criterion in Section 3.4 relies on a calibrated threshold $Z_{\text{thr}}$ derived from normal data, it is important to verify that the method remains stable when calibration data are partially contaminated. We probe the stability of the calibrated threshold $Z_{\text{thr}}$ against $\varepsilon$-contamination, where an $\varepsilon$ fraction of normal calibration windows are replaced by anomalous ones. For each $\varepsilon \in \{0, 3, 5, 10\}$ we recompute $Z_{\text{thr}}^{\varepsilon}$ and report the relative shift $\Delta Z = (Z_{\text{thr}}^{\varepsilon} - Z_{\text{thr}})/Z_{\text{thr}}$ and the resulting point-wise F1 on the test set (no re-tuning of $p$, $\rho$).

Table 3: Effect of calibration contamination on threshold stability and detection performance on SMAP (S-1).

| $\varepsilon$ (%) | $Z_{\text{thr}}$ | $\Delta Z$ (%) | F1 | $\Delta$F1 (pp) |
|---|---|---|---|---|
| 0 | 0.5982 | —— | 0.8095 | —— |
| 3 | 0.6105 | +2.05 | 0.7918 | -2.19 |
| 5 | 0.7075 | +18.3 | 0.5366 | -33.7 |
| 10 | 0.7204 | +20.4 | 0.3832 | -52.6 |

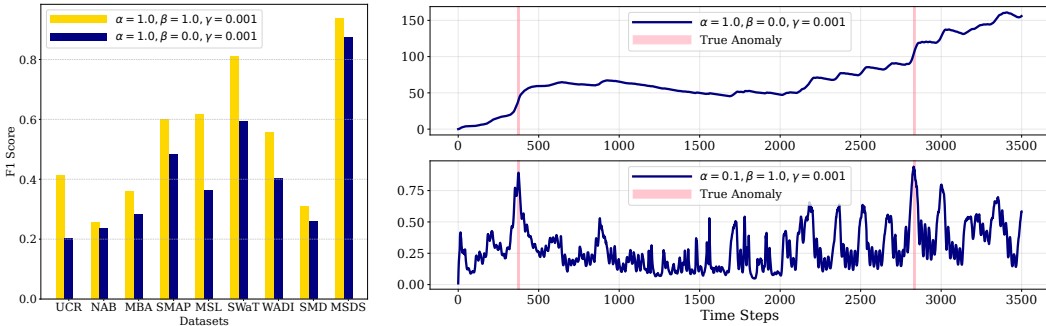

Figure 6: Sensitivity of FOLD to ODE parameters $\alpha, \beta, \gamma$. **(left)** F1-scores across benchmarks for two representative settings. **(right)** Example risk trajectories under different $\beta$ values, showing that $\beta = 1.0$ yields more informative dynamics.

Table 4: Ablation study on stress signal components.

| Method | NAB | | | UCR | | | SMAP | | |
|---|---|---|---|---|---|---|---|---|---|
| | P | R | F1 | P | R | F1 | P | R | F1 |
| NRdetector | 0.0502 | 0.0581 | 0.1032 | 0.6361 | 0.0542 | 0.0908 | 0.6372 | 0.1608 | 0.2367 |
| FOLD (*w/o Uncertainty*) | 0.2083 | 0.0781 | 0.1136 | 0.3417 | 0.4116 | *0.1761* | 0.1879 | 0.0942 | 0.0655 |
| FOLD (*w/o Sensitivity*) | 0.1939 | 0.5937 | *0.2187* | 0.0394 | 0.3502 | 0.0599 | 0.3491 | 0.3532 | *0.2959* |
| FOLD (Full Model) | 0.3519 | 0.6875 | **0.4245** | 0.4267 | 0.4058 | **0.2542** | 0.6820 | 0.7059 | **0.6013** |

Even at $\varepsilon = 10\%$, FOLD retains an F1 of 0.3832 on SMAP (S-1), which, despite the drop from clean calibration, remains substantially higher than strong baselines on the same machine (e.g., NRDetector 0.2032, TranAD 0.0080). Small contaminations ($\varepsilon \leq 3\%$) shift $Z_{thr}$ by only a few percent and lead to modest F1 changes, consistent with the high-quantile stability we exploit.

**Sensitivity to ODE parameters.** We further analyze the role of the coefficients $\alpha, \beta, \gamma$ in the fold-bifurcation inspired ODE (Eq. 7). Figure 6 compares two representative settings: ($\alpha = 1.0, \beta = 1.0, \gamma = 0.001$) versus ($\alpha = 1.0, \beta = 0.0, \gamma = 0.001$). The results highlight the importance of the nonlinear escalation term $-\beta \mathbf{z}(t)^2$: when $\beta = 0$, the risk state $\mathbf{z}(t)$ grows monotonically with accumulated stress, essentially acting as a simple integrator; in contrast, $\beta = 1.0$ introduces nonlinear suppression that yields a more structured and interpretable trajectory, allowing $\mathbf{z}(t)$ (or $\mathbf{z}_{sys}(t)$ for multivariate settings) to capture meaningful rises and falls around anomalous intervals. Across benchmarks (left), $\beta = 1.0$ consistently leads to higher F1-scores, and the temporal plots (right) confirm that the dynamics with $\beta = 1.0$ produce more discriminative risk states. This demonstrates that FOLD's performance does not hinge on delicate parameter tuning, but that incorporating nonlinear escalation ($\beta = 1.0$) is key to realizing the full benefit of fold-bifurcation dynamics. (In Appendix E.2, we report various sensitivity studies on $\alpha, \beta, \gamma$.)

## 6 CONCLUSION

We presented **FOLD**, a fold-bifurcation–inspired framework for point-wise anomaly detection. Instead of relying on reconstruction or prediction errors, FOLD models how sensitivity- and uncertainty-based stress signals accumulate through a simple ODE to trigger a tipping-point transition. Once a forecasting model is trained on normal data, no anomaly labels or additional detector training are required, enabling fully label-free detection. This design provides interpretability grounded in dynamical-systems theory, minimal computational overhead, and consistently strong results under strict point-wise evaluation across 40 benchmarks against 34 state-of-the-art baselines. Beyond accuracy, FOLD yields a transparent continuous risk trajectory via the risk state $\mathbf{z}(t)$, unifying both sudden spikes and gradual drifts within a single mechanism. In future work, we will explore data-adaptive parameterization of the ODE using only normal data and extend FOLD to streaming scenarios with drift-aware calibration.

**Ethics statement.** This work adheres to the ICLR Code of Ethics. Our study uses only publicly available, non-personal time-series benchmarks (e.g., SMAP, MSL, SWaT, WADI, SMD, NAB, UCR) under their respective licenses; no human subjects, personally identifiable information, or protected attributes are involved, and no re-identification is attempted. Data are used solely for research and are processed in accordance with the datasets' terms of use. Because anomaly detection can have dual-use risks (e.g., surveillance or misuse in operational settings), we limit our release to research artifacts (code, configs, and scripts) and provide guidance to avoid deployment on sensitive data without appropriate consent, legal basis, and security review. We disclose no conflicts of interest or external sponsorship that could unduly influence the results.

**Reproducibility Statement.** To ensure the reproducibility and completeness of this paper, we make our code available at `https://github.com/sheoyon-jhin/FOLD`. We give details on our experimental protocol in the Appendix B.

**The Use of Large Language Models (LLMs).** We used ChatGPT as a writing assistant for polishing language and checking notation. No research ideas or content generation were conducted by LLMs.

ACKNOWLEDGMENTS

This work was supported by Institute of Information & communications Technology Planning & Evaluation (IITP) grant funded by the Korea government (MSIT), Development of Financial and Economic Digital Twin Platform based on AI and Data under project number No.2022-0-00857 (34%), and Developing a Sustainable Collaborative Multi-modal Lifelong Learning Framework under project number No.RS-2022-II220113 (33%), and Samsung Research Funding & Incubation Center of Samsung Electronics under project number SRFC-IT2402-08 (33%).

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

Table 5: Notations used in FOLD.

| Symbol | Description | Symbol | Description |
|---|---|---|---|
| $X \in \mathbb{R}^{L \times d}$ | Input sequence | $H$ | Forecast horizon |
| $f_\theta$ | Forecasting model | $\hat{Y} = f_\theta(X)$ | Baseline prediction |
| $P_i$ | $i$-th patch | $X \setminus P_i$ | Input with $P_i$ masked |
| $\hat{Y}_{\setminus i}$ | Perturbed prediction | $D(\cdot, \cdot)$ | Distance function |
| $\delta, \lambda$ | Stress weights | $T_{\mathrm{MC}}$ | MC dropout passes |
| $\epsilon_i$ | Stress score for $P_i$ | $\mathcal{I}(t)$ | Patches covering $t$ |
| $\mathbf{S}(t)$ | Stress signal at $t$ | $\mathbf{z}(t)$ | Risk trajectory |
| $\alpha, \beta, \gamma$ | ODE coefficients | $M_{\mathrm{train}}$ | Maxima set from normal |
| $K$ | Calibration windows | $p$ | Quantile level |
| $\rho$ | Margin for threshold | $Z_{\mathrm{thr}}$ | Risk threshold |
| $\hat{y}(t)$ | Point-wise decision | $\mathbf{z}_{sys}(t)$ | System-level risk score |

## A  ALGORITHMS

---
**Algorithm 1:** FOLD Calibration (normal-only)

---
**Input:** $\mathcal{D}_{\mathrm{train}}^{\mathrm{normal}}, f_\theta, \{P_i\}, \delta, \lambda, (\alpha, \beta, \gamma), h, L, p$
**Output:** $Z_{\mathrm{thr}}$

1 $M_{\mathrm{train}} \leftarrow \varnothing$;
   // Step 1:  Train forecasting model on normal data
2 train $f_\theta$ on $\mathcal{D}_{\mathrm{train}}^{\mathrm{normal}}$;
   // Step 2:  Iterate through each normal training sequence
3 **for** $X \in \mathcal{D}_{train}^{normal}$ **do**
4   $\quad \hat{Y} \leftarrow f_\theta(X)$;
     // Obtain baseline prediction
     // Step 3:  Compute patch-wise perturbations and stress
     signals
5   $\quad$ **for** $P_i$ *in* $X$ **do**
6   $\quad\quad \hat{Y}_{\setminus i} \leftarrow f_\theta(X \setminus P_i)$;
7   $\quad\quad \epsilon_i \leftarrow \delta\, D(\hat{Y}_{\setminus i}, \hat{Y}) + \lambda \left| \mathrm{Var}(\hat{Y}_{\setminus i}) - \mathrm{Var}(\hat{Y}) \right|$;
     // Step 4:  Aggregate patch scores into stress signal
8   $\quad \mathbf{S}(t) \leftarrow \frac{1}{|\mathcal{I}(t)|} \sum_{i \in \mathcal{I}(t)} \epsilon_i$ for $t = 1..L$;
     // Step 5:  Simulate fold-bifurcation ODE
9   $\quad$ simulate Eq. (fold-ode) with input $\mathbf{S}(t)$ to get $\mathbf{z}(t)$;
     // Step 6:  Record maximum risk of this sequence
10  $\quad M_{\mathrm{train}} \leftarrow M_{\mathrm{train}} \cup \{\max_t \mathbf{z}(t)\}$;
   // Step 7:  Define anomaly threshold from high-percentile of
   normal maxima
11 $Z_{\mathrm{thr}} \leftarrow \mathrm{Quantile}_p(M_{\mathrm{train}})$;

---

## B  EXPERIMENTAL ENVIRONMENTS

### B.1  DATASETS

We evaluate FOLD on 40 widely used public benchmarks. Table 6 summarizes their key statistics; the value in parentheses denotes the number of sequences provided by each repository, and all reported scores are averaged over sequences within a dataset.

---

**Algorithm 2:** Point-wise Anomaly Detection(FOLD)

---

**Input:** $X, f_\theta, \{P_i\}, \delta, \lambda, (\alpha, \beta, \gamma), h, L, Z_{\text{thr}}$
**Output:** $\mathbf{z}(t), \hat{y}(t) = \mathbf{1}[\mathbf{z}(t) > Z_{\text{thr}}]$

1  $\hat{Y} \leftarrow f_\theta(X)$; // Forecast $H$-step outputs from the input sequence $X$
2  **for** *each patch $P_i$ in $X$* **do**
3      $\hat{Y}_{\backslash i} \leftarrow f_\theta(X \backslash P_i)$;            // Perturbed prediction by masking $P_i$
4      $\epsilon_i \leftarrow \delta\, D(\hat{Y}_{\backslash i}, \hat{Y}) + \lambda \left| \text{Var}(\hat{Y}_{\backslash i}) - \text{Var}(\hat{Y}) \right|$;     // Stress signal combining
       sensitivity and uncertainty
5  $\mathbf{S}(t) \leftarrow \frac{1}{|\mathcal{I}(t)|} \sum_{i \in \mathcal{I}(t)} \epsilon_i$ for $t = 1..L$; // Aggregate patch-level stress into
   a time-varying signal
6  simulate Eq. 7 with input $\mathbf{S}(t)$ to obtain $\mathbf{z}(t)$;
   ;        // Risk trajectory evolves under fold-bifurcation dynamics
7  **return** $\mathbf{z}(t)$ and $\hat{y}(t) = \mathbf{1}[\mathbf{z}(t) > Z_{\text{thr}}]$;
   ; // Anomalies are flagged once $\mathbf{z}(t)$ leaves the stable basin $Z_{\text{thr}}$,
   realizingFOLD anomaly detection without labels

---

Table 6: Overview of dataset characteristics for the 40 benchmarks in TSB-AD. The symbol '-' in the second column indicates that the dataset is transformed from a multivariate source. The 'Category' column specifies whether the dataset features point anomalies (P) or sequence anomalies (Seq).

| | Name | # TS Collected | # TS Curated | Avg Dim | Avg TS Len | Avg # Anomaly | Avg Anomaly Len | Anomaly Ratio | Category |
|---|---|---|---|---|---|---|---|---|---|
| **TSB-AD-U** | UCR (Dau et al., 2019) | 250 | 228 | 1 | 67818.7 | 1 | 198.9 | 0.6% | P&Seq |
| | NAB (Ahmad et al., 2017) | 58 | 28 | 1 | 5099.7 | 1.6 | 370.1 | 10.6% | Seq |
| | YAHOO (Laptev et al., 2015) | 367 | 259 | 1 | 1560.2 | 5.5 | 2.5 | 0.6% | P&Seq |
| | IOPS (IOPS.ai) | 58 | 17 | 1 | 72792.3 | 25.6 | 48.7 | 1.3% | Seq |
| | MGAB (Thill et al., 2020) | 10 | 9 | 1 | 97777.8 | 9.7 | 20.0 | 0.2% | Seq |
| | WSD (Zhang et al., 2022) | 210 | 111 | 1 | 17444.5 | 5.1 | 25.4 | 0.6% | Seq |
| | SED (Boniol et al., 2021) | 6 | 3 | 1 | 23332.3 | 14.7 | 64.0 | 4.1% | Seq |
| | TODS (Lai et al., 2021) | 15 | 15 | 1 | 5000.0 | 97.3 | 18.7 | 6.3% | P&Seq |
| | NEK (Si et al., 2024) | 48 | 9 | 1 | 1073.0 | 2.9 | 51.1 | 8.0% | P&Seq |
| | Stock (Tran et al., 2016) | 90 | 20 | 1 | 15000.0 | 1246.9 | 1.1 | 9.4% | P&Seq |
| | Power (Keogh et al., 2007) | 1 | 1 | 1 | 35040.0 | 4 | 750 | 8.5% | Seq |
| | Daphnet (Bachlin et al., 2009) (U) | - | 1 | 1 | 38774.0 | 6 | 384.3 | 5.9% | Seq |
| | CATSv2 (Fleith, 2023) (U) | - | 1 | 1 | 300000.0 | 19.0 | 778.9 | 4.9% | Seq |
| | SWaT (Mathur & Tippenhauer, 2016) (U) | - | 1 | 1 | 419919.0 | 27.0 | 1876.0 | 12.1% | Seq |
| | LTDB (Goldberger et al., 2000) (U) | - | 9 | 1 | 99700.0 | 127.5 | 144.5 | 18.6% | Seq |
| | TAO (tao) (U) | - | 3 | 1 | 10000.0 | 838.7 | 1.1 | 9.4% | P&Seq |
| | Exathlon (Jacob et al., 2021) (U) | - | 32 | 1 | 44075.8 | 3.1 | 1577.3 | 11.0% | Seq |
| | MITDB (Goldberger et al., 2000) (U) | - | 8 | 1 | 631250.0 | 68.7 | 451.9 | 4.2% | Seq |
| | MSL (Hundman et al., 2018) (U) | - | 9 | 1 | 3492.0 | 1.3 | 130.0 | 5.8% | Seq |
| | SMAP (Hundman et al., 2018) (U) | - | 19 | 1 | 7700.2 | 1.2 | 210.1 | 2.8% | Seq |
| | SMD (Su et al., 2019) (U) | - | 38 | 1 | 24207.7 | 2.4 | 173.7 | 2.0% | Seq |
| | SVDB (Greenwald, 1990) (U) | - | 20 | 1 | 171380.0 | 36.4 | 292.5 | 3.6% | Seq |
| | OPP(Roggen et al., 2010) (U) | - | 29 | 1 | 16544.8 | 1.4 | 653.4 | 6.4% | Seq |
| **TSB-AD-M** | GHL (Filonov et al., 2016) | 48 | 25 | 19 | 199001.0 | 2.2 | 1035.2 | 1.1% | Seq |
| | Daphnet (Bachlin et al., 2009) | 17 | 1 | 9 | 38774.0 | 6.0 | 384.3 | 5.9% | Seq |
| | Exathlon (Jacob et al., 2021) | 72 | 27 | 21 | 60878.4 | 4.3 | 1373.3 | 9.8% | Seq |
| | Genesis (von Birgelen & Niggemann, 2018) | 1 | 1 | 18 | 16220.0 | 3.0 | 16.7 | 0.3% | Seq |
| | OPP (Roggen et al., 2010) | 24 | 8 | 248 | 17426.75 | 1.4 | 394.3 | 4.1% | Seq |
| | SMD (Su et al., 2019) | 28 | 22 | 38 | 25466.4 | 8.9 | 112.8 | 3.8% | Seq |
| | SWaT (Mathur & Tippenhauer, 2016) | 4 | 2 | 59 | 207457.5 | 16.5 | 1093.6 | 12.7% | Seq |
| | PSM (Abdulaal et al., 2021) | 1 | 1 | 25 | 217624.0 | 72.0 | 338.6 | 11.2% | P&Seq |
| | SMAP (Hundman et al., 2018) | 54 | 27 | 25 | 7855.9 | 1.3 | 196.3 | 2.9% | Seq |
| | MSL (Hundman et al., 2018) | 27 | 16 | 55 | 3119.4 | 1.3 | 111.7 | 5.1% | Seq |
| | CreditCard (Sharafaldin et al., 2018) | 1 | 1 | 29 | 284807.0 | 465.0 | 1.1 | 0.2% | P&Seq |
| | GECCO (Moritz et al., 2018) | 1 | 1 | 9 | 138521.0 | 51.0 | 33.8 | 1.2% | Seq |
| | MITDB (Goldberger et al., 2000) | 48 | 13 | 2 | 336153.8 | 15.2 | 1846.8 | 2.7% | Seq |
| | SVDB (Greenwald, 1990) | 78 | 31 | 2 | 207122.6 | 68.3 | 268.2 | 4.8% | Seq |
| | LTDB (Goldberger et al., 2000) | 7 | 5 | 2 | 100000.0 | 105.0 | 134.4 | 15.5% | Seq |
| | CATSv2 (Fleith, 2023) | 10 | 6 | 17 | 240000.0 | 11.5 | 811.6 | 3.7% | Seq |
| | TAO (tao) | 45 | 13 | 3 | 10000.0 | 788.2 | 1.1 | 8.7% | P&Seq |

### B.1.1 BENCHMARK PROTOCOL: TSB-AD

We adhere to the standard evaluation protocol of the **TSB-AD benchmark** (Liu & Paparrizos, 2024), which features a comprehensive collection of 1,070 high-quality time series curated from 40 diverse datasets. The benchmark is structured into two primary tracks: **Univariate (TSB-AD-U)** and **Multivariate (TSB-AD-M)**.

Consistent with the official leaderboard guidelines, the data is partitioned into three distinct subsets to ensure rigorous and fair evaluation:

- **Evaluation Data:** This set comprises 350 univariate and 180 multivariate time series reserved strictly for testing. The average anomaly ratios are 4.5% and 5.0%, respectively. We report the final performance metrics (VUS-PR, F1) on this split.

- **Training Data:** A short, anomaly-free historical segment is provided for each evaluation series. In our framework, we utilize this segment to train the forecasting backbone ($f_\theta$) and to calibrate the threshold $Z_{thr}$ using the distribution of normal risk scores.

- **Tuning Data:** A separate set of time series (48 for univariate, 20 for multivariate), distinct from the evaluation set, is provided for hyperparameter optimization (HPO). We utilize this split to select optimal hyperparameters (e.g., patch size $P$, margin $\rho$) without leaking information from the test set.

## B.2 SETTINGS

1. GPU: NVIDIA A6000
2. OS: Ubuntu 20.04
3. Framework: PyTorch 2.1.1+cu121
4. CUDA: nvcc V12.1.66 (build cuda_12.1.r12.1/compiler.32415258_0)

## B.3 HYPERPARAMETERS

**Hyperparameter search ranges.** We conducted a grid search over the following ranges:

- **Forecasting models:**
  - Sequence length $\in \{96, 48\}$,
  - Output length $\in \{12, 24\}$,
  - Learning rate $\in \{0.001, 0.005, 0.0001\}$
- **Stress signal calculation:**
  - $D(\cdot, \cdot) \in \{\text{cosine similarity, MSE, MAE}\}$,
  - $\delta \in \{0.5, 1.0, 1.5, 2.0\}$,
  - $\lambda \in \{0.5, 1.0, 1.5, 2.0\}$
  - $T_{MC} \in \{30, 50\}$
- **Fold-bifurcation ODE:**
  - $\alpha \in \{0.5, 1.0, 1.5, 2.0, 2.5\}$,
  - $\beta \in \{0.1, 0.3, 0.5, 0.7, 0.9\}$,
  - $\gamma \in \{0.001, 0.01, 0.05, 0.1, 0.15, 0.2\}$,
  - Number of patches $\in \{6, 8\}$

**Best hyperparameters.** The same best configuration was applied consistently across all datasets:

- **Forecasting models:**
  - Sequence length $= 48$,
  - Output length $= 12$,
  - Learning rate $= 0.005$
- **Stress signal calculation:**
  - $D(\cdot, \cdot) = \text{cosine similarity}$,
  - $\delta = 1.0$,
  - $\lambda = 1.0$
  - $T_{MC} \in \{50\}$
- **Fold-bifurcation ODE:**

- $\alpha = 1.0$,
- $\beta = 0.5$,
- $\gamma = 0.001$,
- Number of patches $= 6$

## B.4 Point-wise evaluation

We reuse TranAD (Tuli et al., 2022) splits/preprocessing and re-score all methods under a unified point-wise protocol. Labels are per timestep; decisions are $\hat{y}(t) = \Bbbk\{\text{score}(t) > \text{threshold}\}$, and Precision/Recall/F1 are computed *over timesteps* (no anomaly-range dilation, no delay tolerance, no smoothing).

**Baselines.** For each baseline, we follow the thresholding strategy originally used in the respective paper (e.g., reconstruction- or prediction-error based detectors use their default criteria). We did not re-tune thresholds, ensuring that FOLD is compared against baselines under their standard settings.

**FOLD.** $\text{score}(t) = \mathbf{z}(t)$; the threshold $Z_{\text{thr}}$ is calibrated from normal training windows as in Sec. 3.4; prediction is $\hat{y}(t) = \Bbbk\{\mathbf{z}(t) > Z_{\text{thr}}\}$.

**Point-wise micro F1 score.** We evaluate at the timestep level with no temporal tolerance ($k=0$). For each time $t$, let $y_{\mathbf{S}}(t) \in \{0, 1\}$ be the ground-truth label and $\hat{y}(t) \in \{0, 1\}$ be the binarized prediction (obtained by thresholding the anomaly score).

Dataset-level counts are aggregated over all $s, t$:

$$\text{TP} = \sum_t \Bbbk[\hat{y}(t)=1 \wedge y(t)=1], \tag{11}$$

$$\text{FP} = \sum_t \Bbbk[\hat{y}(t)=1 \wedge y(t)=0], \tag{12}$$

$$\text{FN} = \sum_t \Bbbk[\hat{y}(t)=0 \wedge y(t)=1]. \tag{13}$$

Micro-precision and recall are $P = \frac{\text{TP}}{\text{TP}+\text{FP}}$ and $R = \frac{\text{TP}}{\text{TP}+\text{FN}}$, and the point-wise micro F1 is

$$\text{F1} = \frac{2PR}{P + R}. \tag{14}$$

No dilation/collar or range merging is applied; a prediction is counted as correct only when it matches the exact timestep ($\hat{y}(t)=y(t)=1$). Unless noted otherwise, all F1 scores reported in this paper use this definition.

**Multivariate setting and interactions.** Let $X \in \mathbb{R}^{L \times D}$ denote a window of $D$ variables. The forecaster $f_\theta$ is multivariate (PatchTST, TimeMixer with cross-feature mixing), hence its prediction $\hat{Y} = f_\theta(X) \in \mathbb{R}^{H \times D}$ already encodes cross-variable dependencies. We compute patch scores $\{\varepsilon_i\}$ by locally masking $X$ with patch $m_i$ and measuring the change of a joint multivariate loss, e.g.,

$$\varepsilon_i = \underbrace{\|\hat{Y} - f_\theta(X \odot m_i)\|_{1, \text{time} \times \text{feat}}}_{\text{sensitivity}} + \lambda \cdot \underbrace{\left(\text{Var}[\hat{Y}] - \text{Var}[f_\theta(X \odot m_i)]\right)}_{\text{uncertainty}},$$

where the loss aggregates over all $D$ features. Because $f_\theta$ is multivariate, masking feature $j$ at time $t$ can change predictions of many other features, and this cross-effect is reflected in $\varepsilon_i$. Patch scores are aligned to time and feature axes to form feature-wise stresses $\mathbf{S}_k(t)$ and their aggregation $\mathbf{S}(t)$:

$$\mathbf{S}_k(t) = \sum_i \varepsilon_i \mathbf{1}[(t, k) \in \text{span}(m_i)], \qquad \mathbf{S}(t) = \sum_{k=1}^{D} w_k \mathbf{S}_k(t) \quad (w_k \geq 0, \sum_k w_k = 1).$$

We then evolve feature-wise risk dynamics

$$\dot{\mathbf{z}}_k(t) = \alpha, \mathbf{S}_k(t) - \beta, \mathbf{z}_k(t)^2 - \gamma \mathbf{z}_k(t),$$

yielding $\mathbf{z}(t) = (z_1(t), \ldots, z_D(t)) \in \mathbb{R}^D$. For readability, figures report a system-level risk obtained by summation, e.g. $\mathbf{z}_{\text{sys}}(t) = \sum_k \mathbf{z}_k(t)$. Anomalies can be declared either when $z_{\text{sys}}(t)$ crosses a calibrated threshold.

### B.5 IMPLEMENTATION DETAILS ABOUT FOLD(CHRONOS)

To demonstrate the model-agnostic nature of FOLD, we replaced the trainable forecasting backbone ($f_\theta$) with Chronos (Ansari et al., 2024), a pre-trained probabilistic time-series foundation model, in a zero-shot setting. **Specifically, we utilized the `amazon/chronos-t5-small` variant (approx. 46M parameters) to evaluate the framework's capability even with a lightweight foundation model.** Since Chronos natively outputs a predictive distribution rather than a point estimate, we adapt the stress signal calculation (Eq. 4) as follows:

**Sensitivity Term.** Instead of retraining or fine-tuning, we use the pre-trained Chronos model to generate the median forecast $\hat{Y}_{median}$. The sensitivity is computed as the deviation of this median forecast under local masking:

$$D(\hat{Y}_{\setminus i}, \hat{Y}) = \|\hat{Y}_{median} - \hat{Y}_{\setminus i,median}\| \tag{15}$$

**Uncertainty Term.** Unlike standard deep learning models that require MC Dropout for uncertainty estimation, Chronos provides a distribution of sample trajectories. We directly utilize the variance of these generated samples as the uncertainty measure:

$$\text{Uncertainty} = |Var(\hat{Y}_{\setminus i}) - Var(\hat{Y})| \tag{16}$$

where $Var(\cdot)$ denotes the variance across the sampled forecasts from Chronos.

This adaptation allows FOLD to leverage the superior zero-shot generalization of foundation models without any additional training or architectural modifications. The quantitative results of this integration are presented in Table 1.

## C ADDITIONAL EXPERIMENTAL RESULTS

We follow TranAD (Tuli et al., 2022) for splits and preprocessing (native dimensionality: NAB/UCR univariate; others multivariate) and re-evaluate all methods under a unified point-wise protocol: for each timestep $t$, $\hat{y}(t) = \mathbb{1}\{\mathbf{z}(t) > Z_{\text{thr}}\}$ and Precision/Recall/F1 are computed over timesteps (no dilation, no delay tolerance, no smoothing). This stricter protocol replaces window-level scoring common in prior work. See Appendix B for dataset, hyperparameter, and evaluation details.

### C.1 EXPERIMENTAL RESULTS

Table 7 reports the performance of FOLD compared with 10 state-of-the-art baselines across 9 benchmarks. Under the point-wise anomaly detection setting, which demands anomalies to be identified at the exact timestep, many prior methods show degraded performance, suggesting that earlier window-based results may have overstated fine-grained accuracy.

Despite this stricter setting, FOLD achieves consistently strong F1-scores across domains. On datasets dominated by sudden deviations (e.g., NAB, UCR), FOLD matches or surpasses prediction-based approaches. On benchmarks characterized by gradual stress accumulation (e.g., SMAP, MSL), FOLD further outperforms prediction- and distance-based methods. Notably, on NAB and SMAP, FOLD improves over the strongest baselines by 216.4% and 170.1%, respectively, highlighting its robustness across both low-resource and high-performing settings.

Another key finding is that FOLD maintains competitive performance regardless of the forecasting backbone. Substituting PatchTST with simpler models such as DLinear or TimeMixer yields comparable results, indicating that FOLD does not hinge on forecaster sophistication. Instead, the strength of FOLD lies in transforming sensitivity and uncertainty signals into a dynamical stress process via fold-bifurcation modeling. This demonstrates that it is not merely the presence of error signals but their principled accumulation and state-transition modeling that drive accurate anomaly detection.

We also compare with NRDetector (Wang et al., 2025), a recent method explicitly designed for point-wise anomaly detection. While NRDetector narrows the evaluation gap by aligning its design with stricter metrics, FOLD consistently outperforms it across multiple benchmarks. This advantage arises from our dynamical-systems formulation, which unifies sudden and gradual stresses under a

Table 7: Performance comparison of FOLD with baseline methods across all datasets. P: Precision, R: Recall, F1: F1-score. The best F1-scores are shown in **bold** , and the second-best in *italic* . All results are averaged over 3 independent runs standard deviations are reported in Appendix D.

| Method | NAB | | | UCR | | | MBA | | |
|---|---|---|---|---|---|---|---|---|---|
| | P | R | F1 | P | R | F1 | P | R | F1 |
| LSTM-AD | 0.0501 | 0.2917 | 0.0855 | 0.2083 | 0.0113 | 0.0214 | 0.5822 | 0.2234 | 0.3229 |
| OmniAnomaly | 0.0757 | 0.3073 | 0.1215 | 0.0161 | 0.3125 | 0.0306 | 0.5045 | 0.1712 | 0.2556 |
| MSCRED | 0.0616 | 0.2187 | 0.0961 | 0.0053 | 0.0045 | 0.0049 | 0.7414 | 0.2261 | 0.3465 |
| MAD-GAN | 0.0807 | 0.3958 | 0.1341 | 0.1609 | 0.1863 | 0.1727 | 0.7018 | 0.2299 | 0.3463 |
| USAD | 0.0843 | 0.6719 | 0.1498 | 0.0412 | 0.0824 | 0.0549 | 0.6565 | 0.2234 | 0.3333 |
| MTAD-GAT | 0.0571 | 0.1875 | 0.0875 | 0.0625 | 0.0039 | 0.0073 | 0.6313 | 0.1876 | 0.2892 |
| GDN | 0.0617 | 0.2188 | 0.0963 | 0.1250 | 0.1250 | 0.1250 | 0.4607 | 0.1577 | 0.2349 |
| Anomaly Transformer | 0.0591 | 0.5000 | 0.1057 | 0.0303 | 0.1666 | 0.0513 | 0.2487 | 0.0285 | 0.0511 |
| TranAD | 0.0746 | 0.3437 | 0.1226 | 0.1333 | 0.0180 | 0.0317 | 0.7478 | 0.1334 | 0.2264 |
| NRdetector | 0.0502 | 0.0581 | 0.0539 | 0.6361 | 0.0542 | 0.0999 | 0.5000 | 0.3920 | 0.4394 |
| **FOLD**(DLinear) | 0.3519 | 0.6875 | *0.4655* | 0.4267 | 0.4058 | **0.4160** | 0.7664 | 0.3807 | **0.5087** |
| **FOLD**(PatchTST) | 0.3147 | 0.9600 | **0.4740** | 0.1361 | 1.0000 | 0.2396 | 0.5981 | 0.3218 | *0.4184* |
| **FOLD**(TimeMixer) | 0.2993 | 0.9607 | 0.4564 | 0.2725 | 0.6726 | *0.3879* | 0.4578 | 0.3139 | 0.3724 |

| Method | SMAP | | | MSL | | | SWaT | | |
|---|---|---|---|---|---|---|---|---|---|
| | P | R | F1 | P | R | F1 | P | R | F1 |
| LSTM-AD | 0.1197 | 0.2969 | 0.1706 | 0.0392 | 0.3351 | 0.0702 | 0.7778 | 0.0108 | 0.0213 |
| OmniAnomaly | 0.0580 | 0.1762 | 0.0873 | 0.0424 | 0.3189 | 0.0748 | 0.9742 | 0.6475 | 0.7779 |
| MSCRED | 0.0928 | 0.0085 | 0.0156 | 0.0538 | 0.2665 | 0.0895 | 1.0000 | 0.3897 | 0.5608 |
| MAD-GAN | 0.0682 | 0.1910 | 0.1005 | 0.0217 | 0.0197 | 0.0207 | 0.9243 | 0.3602 | 0.5183 |
| USAD | 0.0743 | 0.2831 | 0.1177 | 0.0393 | 0.3209 | 0.0700 | 0.9870 | 0.4721 | 0.6387 |
| MTAD-GAT | 0.0170 | 0.7429 | 0.0332 | 0.1453 | 0.3556 | 0.2063 | 0.9731 | 0.6195 | 0.7570 |
| GDN | 0.4056 | 0.0495 | 0.0882 | 0.0602 | 0.3298 | 0.1018 | 0.9673 | 0.6428 | 0.7723 |
| Anomaly Transformer | 0.3195 | 0.0255 | 0.0472 | 0.1557 | 0.0159 | 0.0289 | 0.4663 | 0.1514 | 0.2285 |
| TranAD | 0.1185 | 0.2197 | 0.1540 | 0.2326 | 0.0323 | 0.0567 | 0.9964 | 0.4363 | 0.6068 |
| NRdetector | 0.6372 | 0.1608 | 0.2568 | 0.4884 | 0.1511 | 0.2308 | 0.7336 | 0.5772 | 0.6460 |
| **FOLD**(DLinear) | 0.6820 | 0.7059 | **0.6937** | 0.3514 | 0.5821 | 0.4382 | 0.6824 | 1.0000 | **0.8112** |
| **FOLD**(PatchTST) | 0.7238 | 0.5214 | *0.6062* | 0.5429 | 0.6727 | **0.6009** | 0.8212 | 0.7882 | *0.8043* |
| **FOLD**(TimeMixer) | 0.5103 | 0.6928 | 0.5877 | 0.5956 | 0.5117 | *0.5505* | 0.7804 | 0.8046 | 0.7923 |

| Method | WADI | | | SMD | | | MSDS | | |
|---|---|---|---|---|---|---|---|---|---|
| | P | R | F1 | P | R | F1 | P | R | F1 |
| LSTM-AD | 0.0472 | 0.0012 | 0.0023 | 0.0389 | 0.0031 | 0.0057 | 0.0075 | 0.2692 | 0.0145 |
| OmniAnomaly | 0.1249 | 0.0275 | 0.0451 | 0.4779 | 0.0543 | 0.0975 | 0.8414 | 0.7997 | 0.8200 |
| MSCRED | 0.0519 | 0.2069 | 0.0830 | 0.9783 | 0.0921 | 0.1684 | 0.4324 | 0.0075 | 0.0147 |
| MAD-GAN | 0.0069 | 0.0852 | 0.0128 | 0.9999 | 0.0064 | 0.0127 | 1.0000 | 0.1542 | 0.2672 |
| USAD | 0.0969 | 0.3584 | 0.1526 | 0.6320 | 0.1064 | 0.1821 | 0.7263 | 0.9999 | 0.8414 |
| MTAD-GAT | 0.0846 | 0.2894 | 0.1309 | 0.4803 | 0.0879 | 0.1486 | 0.0282 | 0.5962 | 0.0538 |
| GDN | 0.0820 | 0.2256 | 0.1203 | 0.4658 | 0.1277 | 0.2004 | 0.0004 | 0.9999 | 0.0008 |
| Anomaly Transformer | 0.1428 | 0.6571 | 0.2346 | 0.1698 | 0.0362 | 0.0597 | 0.0374 | 0.5103 | 0.0696 |
| TranAD | 0.2157 | 0.3433 | 0.2649 | 0.6536 | 0.0660 | 0.1199 | 0.7264 | 0.9999 | 0.8414 |
| NRdetector | 0.4849 | 0.2712 | 0.3479 | 0.2232 | 0.0722 | 0.1091 | 0.4943 | 1.0000 | 0.6615 |
| **FOLD**(DLinear) | 0.5299 | 0.7037 | **0.6046** | 0.3095 | 0.4258 | **0.3585** | 0.8531 | 0.8822 | *0.8674* |
| **FOLD**(PatchTST) | 0.4144 | 0.6731 | *0.5130* | 0.4151 | 0.3023 | 0.3498 | 0.8812 | 0.8781 | **0.8796** |
| **FOLD**(TimeMixer) | 0.4065 | 0.6900 | 0.5116 | 0.4224 | 0.3095 | *0.3572* | 0.8023 | 0.8692 | 0.8344 |

single framework. Thus, FOLD not only competes strongly with state-of-the-art baselines but also advances point-wise anomaly detection beyond current specialized methods.

# D  EXPERIMENTS RESULTS WITH STANDARD DEVIATION

The reported mean and standard deviation are computed by repeating the forecasting model training three times with different random seeds. Since the anomaly detection pipeline builds on the trained forecaster, the reported deviations reflect both variability in the forecasting stage and its propagation into stress-signal based detection.

Table 8: FOLD anomaly detection results with mean and standard deviation over three independent runs of the forecasting model.

| Method | NAB | UCR | MBA |
|---|---|---|---|
| | F1 | F1 | F1 |
| **FOLD**(DLinear) | $0.4655 \pm 0.000$ | $0.4160 \pm 0.008$ | $0.5087 \pm 0.001$ |
| **FOLD**(PatchTST) | $0.4740 \pm 0.003$ | $0.2396 \pm 0.007$ | $0.3018 \pm 0.007$ |
| **FOLD**(TimeMixer) | $0.4564 \pm 0.001$ | $0.3879 \pm 0.001$ | $0.2072 \pm 0.005$ |
| Method | SMAP | MSL | SWaT |
| | F1 | F1 | F1 |
| **FOLD**(DLinear) | $0.6937 \pm 0.002$ | $0.4382 \pm 0.001$ | $0.8112 \pm 0.005$ |
| **FOLD**(PatchTST) | $0.6062 \pm 0.010$ | $0.6009 \pm 0.000$ | $0.8043 \pm 0.002$ |
| **FOLD**(TimeMixer) | $0.5877 \pm 0.012$ | $0.5505 \pm 0.002$ | $0.7923 \pm 0.005$ |
| Method | WADI | SMD | MSDS |
| | F1 | F1 | F1 |
| **FOLD**(DLinear) | $0.6046 \pm 0.001$ | $0.3585 \pm 0.004$ | $0.8674 \pm 0.001$ |
| **FOLD**(PatchTST) | $0.5130 \pm 0.001$ | $0.3498 \pm 0.005$ | $0.8796 \pm 0.010$ |
| **FOLD**(TimeMixer) | $0.5116 \pm 0.002$ | $0.3572 \pm 0.002$ | $0.8344 \pm 0.012$ |

# E ADDITIONAL ABLATION

## E.1 CHOICE OF DISTANCE METRIC FOR SENSITIVITY TERM

To evaluate the impact of different distance metrics in the sensitivity term, we conducted an ablation study comparing MSE, MAE and cosine similarity as the function $D(\cdot, \cdot)$. All other components of FOLD (DLinear) were kept fixed. As shown in Table 9, cosine similarity consistently improvements. These results confirm that FOLD is robust to the choice of distance metric, with cosine similarity selected as the default due to its stable performance across benchmarks.

Table 9: Ablation study on distance metric for sensitivity term.

| Method | NAB | | | MSL | | | SMD | | |
|---|---|---|---|---|---|---|---|---|---|
| | P | R | F1 | P | R | F1 | P | R | F1 |
| FOLD (MSE) | 0.2984 | 0.9351 | *0.4524* | 0.3409 | 0.5901 | *0.4322* | 0.2942 | 0.3087 | *0.3013* |
| FOLD (MAE) | 0.2812 | 0.8426 | 0.4216 | 0.3381 | 0.5206 | 0.4099 | 0.3112 | 0.2724 | 0.2905 |
| FOLD (Cosine similarity) | 0.2993 | 0.9607 | **0.4564** | 0.3514 | 0.5821 | **0.4382** | 0.3095 | 0.4258 | **0.3585** |

## E.2 SENSITIVITY TO ODE PARAMETERS

We analyze the sensitivity of FOLD to the coefficients $\alpha, \beta, \gamma$ in the fold-bifurcation inspired ODE (Eq. 7). Each parameter is varied individually while the others are fixed, and the resulting F1-scores are reported across representative benchmarks. As shown in Figure 7, FOLD maintains stable performance over a wide range of values, indicating robustness to hyperparameter choices. Only extreme values lead to noticeable degradation, underscoring the importance of avoiding pathological settings rather than requiring precise tuning.

## E.3 SENSITIVITY TO THE NUMBER OF PATCHES

We further analyze the effect of the patch number $N$, which controls the temporal granularity of stress signal estimation. As shown in Figure 8, FOLD maintains consistently superior performance compared to baselines across a wide range of $N$. While moderate patch sizes (e.g., $N = 6$–$10$) yield slightly better results, the overall performance does not degrade significantly even at extreme values. This robustness indicates that FOLD does not rely on a finely tuned patch size: the forecasting

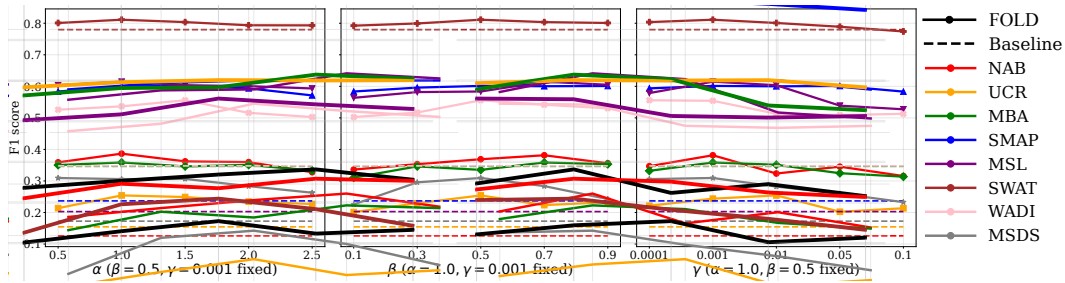

Figure 7: Sensitivity of FOLD to ODE parameters $\alpha, \beta, \gamma$. Performance is stable across broad ranges.

backbone provides sufficiently stable sensitivity and uncertainty signals, and the fold-bifurcation dynamics remain effective regardless of the precise partitioning. Therefore, FOLD's advantage stems from its principled stress accumulation modeling rather than from sensitive hyperparameter choices.

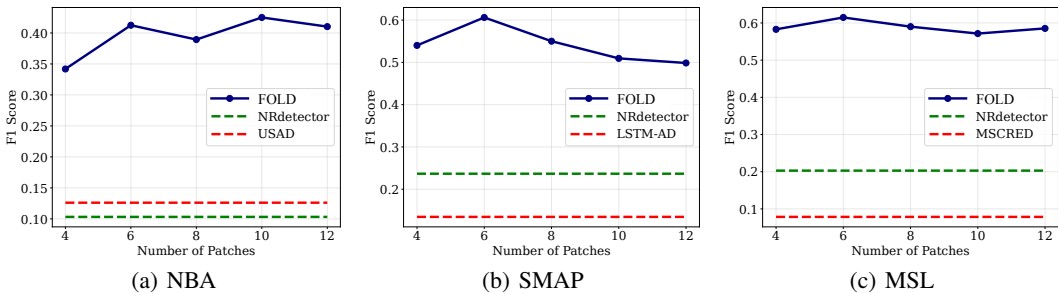

(a) NBA          (b) SMAP          (c) MSL

Figure 8: Sensitivity study on the number of patches $N$. We report F1-scores of FOLD compared with representative baselines on NAB, SMAP, and MSL. Across all datasets, FOLD consistently outperforms the baselines, and its performance remains stable across a broad range of $N$, demonstrating robustness to patch partitioning choices.

### E.4 THRESHOLD ROBUSTNESS ANALYSIS

To complement the theoretical discussion in Section 3.4, we empirically assess the robustness of the threshold $Z_{\text{thr}}$.

**Bootstrap confidence intervals.** For each dataset, we resample the calibration set $M_{\text{train}}$ ($B = 1000$ replicates) and recompute $Z_{\text{thr}} = \text{Quantile}_p(M_{\text{train}})$ with $p = 0.99$. Table 10 reports the mean and 95% confidence interval, showing that the variation of $Z_{\text{thr}}$ is below 5% across all datasets.

Table 10: Bootstrap confidence intervals of $Z_{\text{thr}}$ on selected datasets.

| Dataset | $Z_{\text{thr}}$ (mean) | 95% CI |
|---|---|---|
| UCR | 2.518 | [2.502, 2.534] |
| SMAP | 0.598 | [0.590, 0.623] |
| SMD | 0.713 | [0.699, 0.731] |
| MSL | 0.382 | [0.369, 0.401] |

**Calibration robustness.** We probe the stability of the calibrated threshold $Z_{\text{thr}}$ against $\varepsilon$-contamination, where an $\varepsilon$ fraction of normal calibration windows are replaced by anomalous ones. For each $\varepsilon \in \{0, 3, 5, 10\}$ we recompute $Z_{\text{thr}}^{\varepsilon}$ and report the relative shift $\Delta Z = (Z_{\text{thr}}^{\varepsilon} - Z_{\text{thr}})/Z_{\text{thr}}$ and the resulting point-wise F1 on the test set (no re-tuning of $p, \rho$).

Table 11: Effect of calibration contamination on threshold stability and detection performance on SMAP (S-1).

| $\varepsilon$ (%) | $Z_{\text{thr}}$ | $\Delta Z$ (%) | F1 | $\Delta$F1 (pp) |
|---|---|---|---|---|
| 0 | 0.5982 | —— | 0.8095 | —— |
| 3 | 0.6105 | +2.05 | 0.7918 | -2.19 |
| 5 | 0.7075 | +18.3 | 0.5366 | -33.7 |
| 10 | 0.7204 | +20.4 | 0.3832 | -52.6 |

Even at $\varepsilon = 10\%$, FOLD retains an F1 of $0.3832$ on SMAP (S-1), which,despite the drop from clean calibration, remains substantially higher than strong baselines on the same machine (e.g., NRDetector 0.1032, TranAD 0.0080).Small contaminations ($\varepsilon \leq 3\%$) shift $Z_{\text{thr}}$ by only a few percent and lead to modest F1 changes, consistent with the high-quantile stability we exploit.

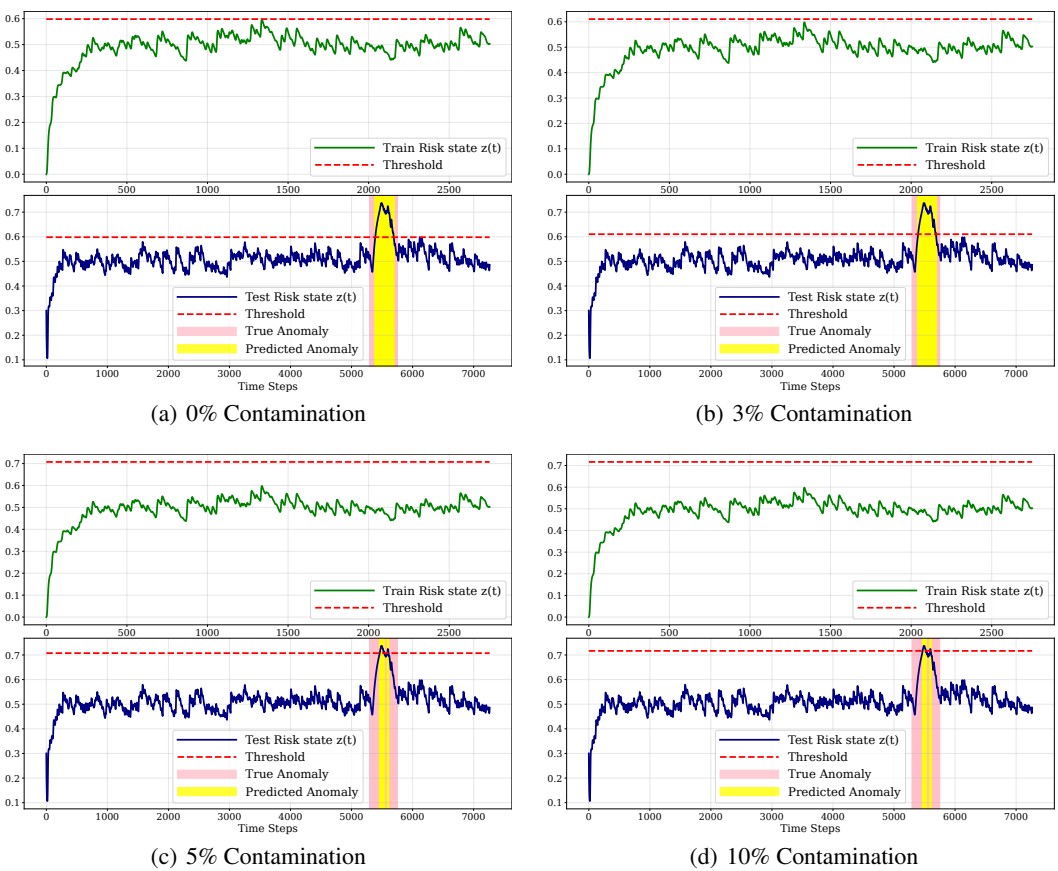

(a) 0% Contamination

(b) 3% Contamination

(c) 5% Contamination

(d) 10% Contamination

Figure 9: Robustness of threshold on SMAP.

## F EFFICIENCY AND COMPLEXITY ANALYSIS

We analyze the computational efficiency of FOLD from both theoretical complexity and empirical resource usage perspectives.

**Theoretical Complexity.** Unlike many SOTA anomaly detectors built on Transformer backbones with quadratic complexity $O(L^2)$ (where $L$ is the sequence length), FOLD utilizes lightweight backbones (e.g., DLinear) with linear complexity $O(L)$. During inference, FOLD computes the stress signal by evaluating $P$ patches. Even with $K$ MC dropout samples for uncertainty estimation, the total scoring complexity is $O(K \cdot P \cdot L)$. Since $K$ and $P$ are small constants independent of $L$, FOLD

Table 12: Computational efficiency comparison on the MSL dataset.

| Model | GPU Memory Usage (MB) | Additional Memory at Inference (MB) | System Memory Usage (MB) |
|---|---|---|---|
| LSTM-AD | 26.73 | 8.180 | 1094.76 |
| OmniAnomaly | 19.73 | 1.320 | 1343.11 |
| MSCRED | 42.87 | 12.54 | 80409.57 |
| MAD-GAN | 29.87 | 1.300 | 1036.21 |
| USAD | 29.92 | 1.200 | 1409.09 |
| MTAD-GAN | 36.87 | 9.872 | 3024.28 |
| GDN | 38.41 | 10.18 | 4886.08 |
| Anomaly Transformer | 28.32 | 351.4 | 3687.15 |
| TranAD | 40.42 | 1.240 | 1016.51 |
| NRdetector | 31.22 | 214.1 | 2965.31 |
| FOLD (DLinear) | **18.04** | **1.170** | **893.49** |

maintains an asymptotic linear complexity $O(L)$, offering a fundamental efficiency advantage over $O(L^2)$ methods. Furthermore, the patch evaluations are independent and massively parallelizable, allowing them to be processed in a single batch on GPUs.

**Empirical Efficiency.** Table 12 compares peak GPU memory, additional memory at inference, and overall system memory on the MSL dataset. FOLD achieves the lowest memory footprint across all metrics. Specifically, it requires only 18.04 MB of GPU memory and minimal additional inference memory (1.17 MB), significantly outperforming complex deep learning baselines like Anomaly Transformer (28.32 MB / 351.4 MB) and MSCRED (42.87 MB). This confirms that FOLD's linear complexity translates directly to practical, real-time efficiency suitable for industrial deployment.

**Practical Implications (Training-Free).** Crucially, the anomaly detector in FOLD is a **training-free extension** of the forecasting model. Unlike other methods that require training a separate detector or fine-tuning on anomaly scores, FOLD is derived directly from the pre-trained forecaster without additional parameter optimization. Combined with the massively parallelizable inference computations (as discussed in Theoretical Complexity), this training-free nature minimizes deployment costs and makes FOLD highly amenable to real-time industrial applications.

## G  ADDITIONAL VISUALIZATION

## H  THEORETICAL ANALYSIS OF FOLD-BIFURCATION DYNAMICS

In this section, we provide a formal derivation demonstrating that the unified detection capability of FOLD is not an ad-hoc design choice but an intrinsic mathematical property of the governing ODE. We analyze the behavior of Eq. (7) under different time-scales of the input stress $S(t)$.

### H.1  UNIFIED MECHANISM VIA TIME-SCALE SEPARATION

The system state $z(t)$ evolves according to the single governing equation:

$$\frac{dz}{dt} = \underbrace{\alpha S(t)}_{\text{Forcing}} - \underbrace{(\beta z^2 + \gamma z)}_{\text{Restoring}} \tag{17}$$

We analyze the dominant terms in this equation based on the time-scale of the input stress relative to the system's intrinsic relaxation time.

**Regime 1: Quasi-Static Limit (Gradual Accumulation).** When the stress $S(t)$ varies slowly ($\frac{dS}{dt} \approx 0$), the time derivative $\frac{dz}{dt}$ becomes negligible compared to the algebraic terms. In this regime,

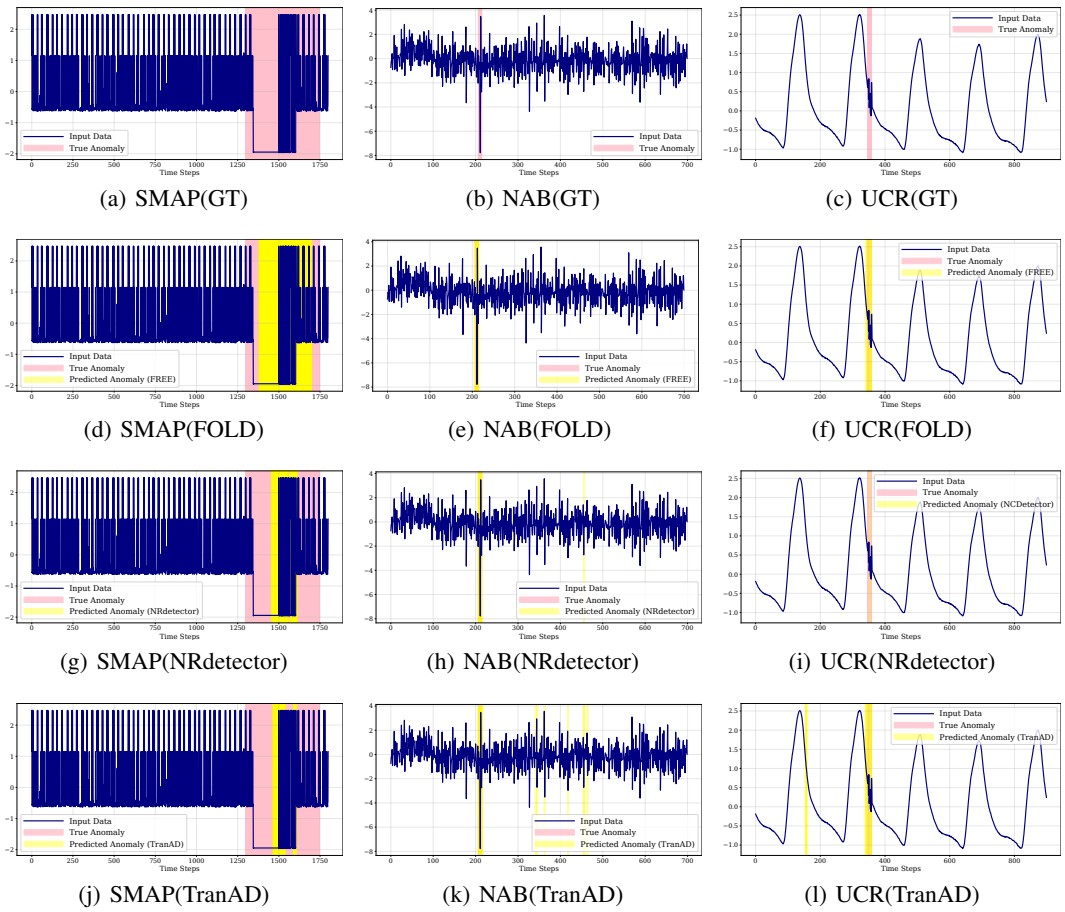

Figure 10: Qualitative comparison on three benchmark datasets representing distinct anomaly types: Gradual (SMAP), Abrupt (NAB), and Periodic (UCR). We compare FOLD against state-of-the-art point-wise (NRDetector) and window-based (TranAD) baselines. FOLD demonstrates robust detection across diverse scenarios, accurately capturing gradual drifts in SMAP and abrupt spikes in NAB, whereas baselines exhibit missed detections or false positives (e.g., NRDetector on NAB, TranAD on SMAP).

the system operates on a *slow manifold*, tracking the instantaneous moving equilibrium where forces balance:

$$\alpha S(t) \approx \beta z(t)^2 + \gamma z(t) \tag{18}$$

Consequently, the risk state $z(t)$ algebraically tracks the magnitude of $S(t)$. Anomaly detection is triggered via a **Saddle-Node Bifurcation** when the stress exceeds the system's capacity to maintain this stable equilibrium. This mathematically explains the detection of gradual drifts (e.g., in SMAP).

**Regime 2: Impulsive Limit (Abrupt Spike).** When the stress $S(t)$ acts as a large-magnitude impulse over a short duration $\Delta t$ (i.e., $S(t) \gg \beta z^2 + \gamma z$), the forcing term dominates the restoring dynamics. The ODE asymptotically simplifies to a pure integrator:

$$\frac{dz}{dt} \approx \alpha S(t) \implies \Delta z \approx \int_t^{t+\Delta t} \alpha S(\tau) d\tau \tag{19}$$

Here, $z(t)$ undergoes an instantaneous state jump proportional to the total accumulated energy of the spike. This explains why the model reacts immediately to abrupt anomalies (e.g., in NAB) without delay.

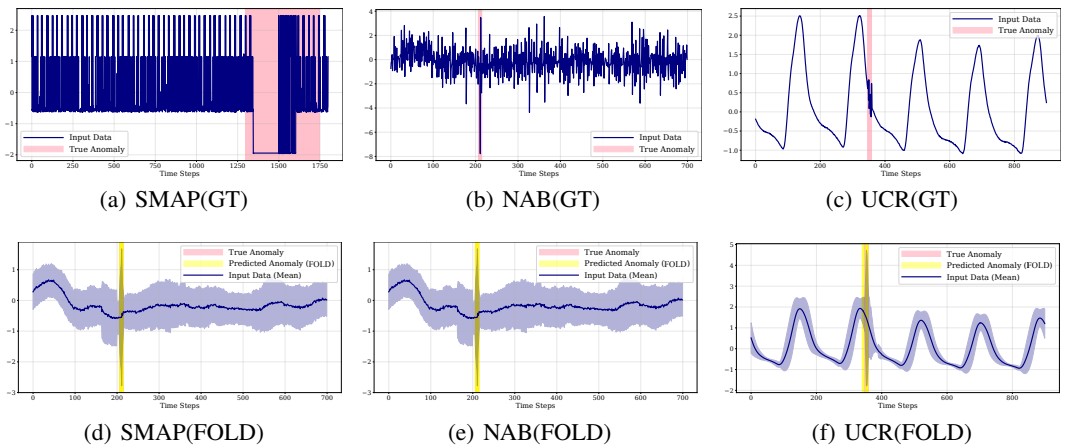

(a) SMAP(GT)          (b) NAB(GT)          (c) UCR(GT)

(d) SMAP(FOLD)          (e) NAB(FOLD)          (f) UCR(FOLD)

Figure 11: Visualization of uncertainty assessments to demonstrate reliability and confidence. The blue line represents the input data, and the shaded region indicates the uncertainty band (derived from MC dropout). Note that the uncertainty band remains narrow during normal states (indicating high confidence) but significantly widens at the onset of anomalies, triggering the risk accumulation mechanism. This visually confirms that FOLD's detections are driven by model-intrinsic risk assessment rather than random fluctuations.

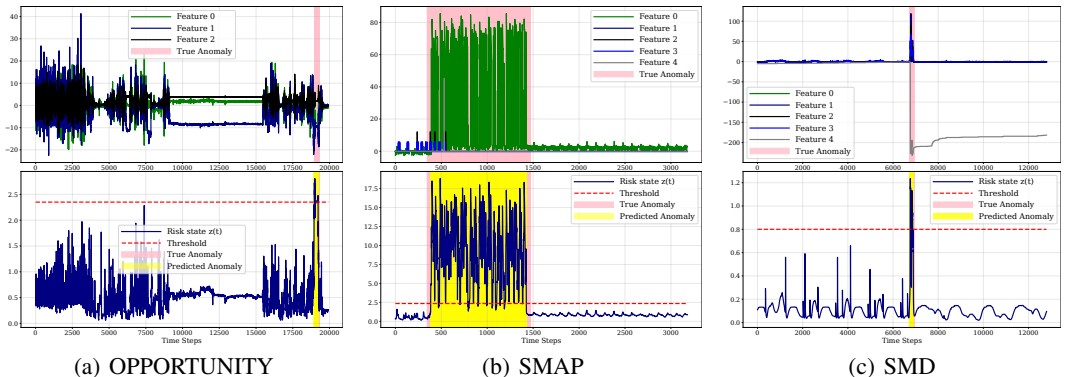

(a) OPPORTUNITY          (b) SMAP          (c) SMD

Figure 12: Visualization of Risk state ($\mathbf{z}_{sys}(t)$) on Multivariate Datasets. Although FOLD computes risk feature-wise, the final decision is based on the aggregated system-level risk (blue line). This plot demonstrates how the aggregated $\mathbf{z}_{sys}(t)$ successfully captures anomalies in high-dimensional datasets (OPPORTUNITY, SMAP, SMD) by integrating stress from multiple features.

## H.2 THEORETICAL GUARANTEE OF ROBUSTNESS

The robustness of FOLD is guaranteed by the **Linear Stability Analysis** of the restoration term. In the absence of strong forcing (i.e., $S(t) \to 0$), the dynamics simplify to $\frac{dz}{dt} = -(\beta z^2 + \gamma z)$. Near the stable equilibrium ($z \approx 0$), the linearized dynamics are governed by the eigenvalue $\lambda \approx -\gamma$. Since $\gamma > 0$, we have $\lambda < 0$, which guarantees **exponential decay** of perturbations:

$$z(t) \propto e^{-\gamma t} \tag{20}$$

This proves that the system is mathematically guaranteed to return to the stable basin after a transient shock, ensuring that the model naturally heals itself and prevents false positives from error propagation.

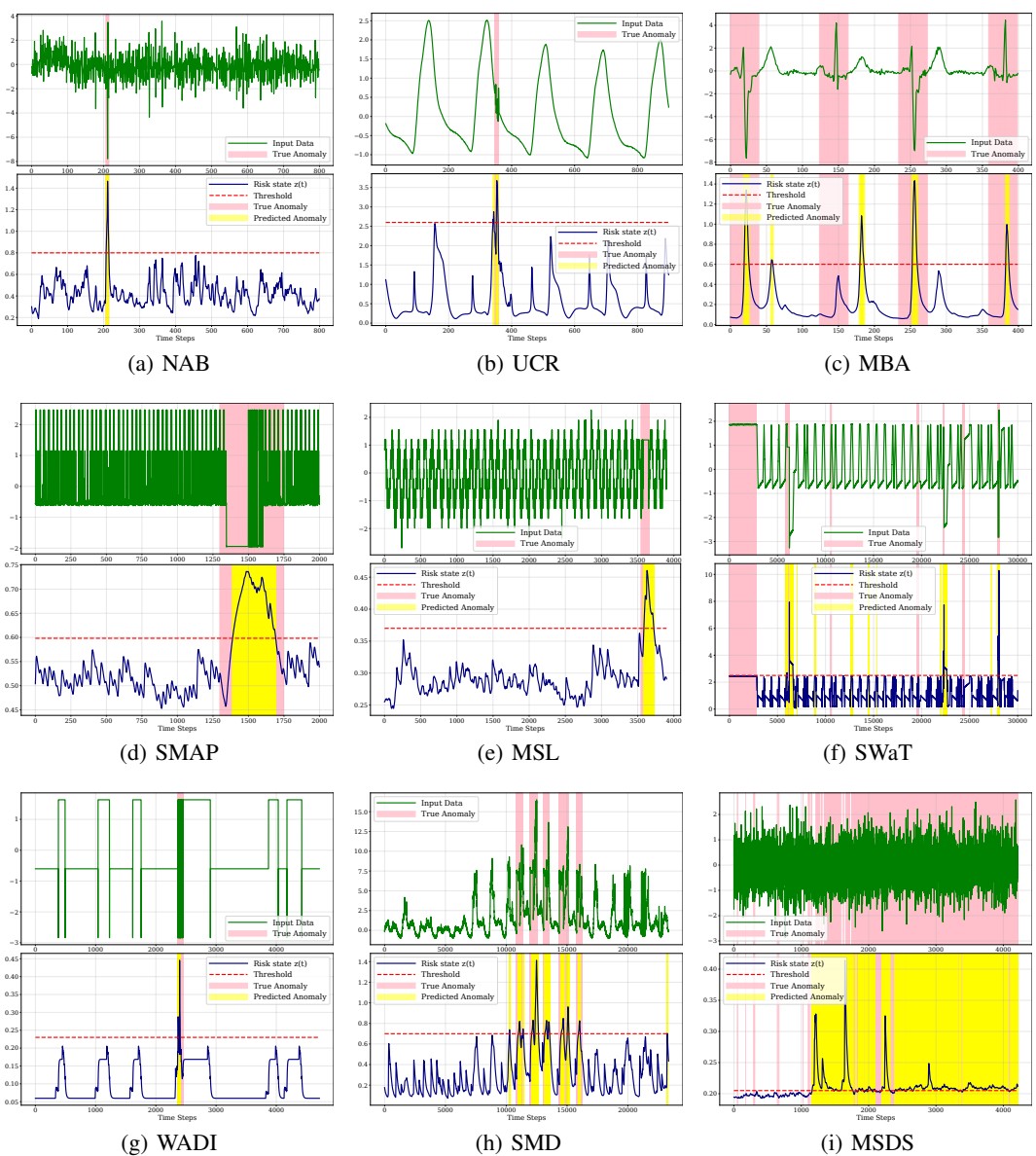

Figure 13: Visualization of anomaly detection results

