# OpenReview forum: "Point-wise Anomaly Detection via Fold-bifurcation ODE"
_ICLR.cc/2026/Conference — ICLR 2026 Poster_

### Official Review · Reviewer_u6SV · 2025-10-27

**Soundness:** 3
**Presentation:** 3
**Contribution:** 2
**Rating:** 6
**Confidence:** 4

**Summary:**

The paper presents FREE (Free point-wise Anomaly Detection via fold-bifurcation), a framework for time-series anomaly detection that targets strict point-wise evaluation rather than coarse window-level metrics. FREE computes stress signals from a forecasting model and feeds them into a fold-bifurcation–inspired ODE to track a system’s risk state; anomalies are flagged when this state crosses a threshold calibrated on normal data.

**Strengths:**

FREE, grounded in fold-bifurcation dynamics, recasts anomaly detection as a point-wise decision process driven by the progression from stress to critical transition, providing strong theoretical footing and inherent interpretability. By integrating forecast-derived stress signals into a bifurcation equation, it captures the buildup of gradual pressures while remaining sensitive to sudden tipping events. Extensive multi-benchmark evaluation demonstrates strong performance—particularly under strict point-wise settings—achieving a balanced mix of accuracy and efficiency and underscoring its practical value.

**Weaknesses:**

1.The central assumption—that many real-world failures stem from gradually accumulating stress leading the system toward a critical transition rather than isolated spikes—seems reasonable, but the paper would benefit from additional concrete scenarios to broaden and clarify its practical applicability.

2.For the fold-bifurcation–inspired ODE modeling, the linkage between theory and specific application domains is underdeveloped; the discussion remains largely theory-driven without fully mapping the formulation to domain variables, constraints, and operational workflows.

3.On visualization, it’s hard to see—at a glance—where the method outperforms alternatives. Clear comparative plots would help, ideally accompanied by visualized uncertainty assessments to show reliability and confidence.

4.Regarding thresholding, the confidence-interval analysis feels insufficient. The large differences in preset thresholds—are they driven by dataset standards? Could thresholds be informed by priors or domain knowledge to improve calibration stability?

5.The efficiency results are compelling, but a deeper analysis is needed to understand the sources of improvement relative to other methods.

**Questions:**

My primary concerns center on actionable guidance from theory to practice, the method’s novelty relative to comparable baselines, a thorough efficiency analysis, and a clear discussion of threshold selection and robustness.

---

> ### Author Response · Authors · 2025-11-21
>
> We sincerely thank Reviewer u6SV for the constructive feedback and the encouraging assessment of our work. We are glad that you recognized our method's 'strong theoretical footing' and 'inherent interpretability.'
>
> We especially appreciate your valuable suggestions on strengthening the link between theory and practice.
>
> ## W1: Clarification of the "Stress Accumulation" Assumption
>
> * We thank both reviewers for their valuable feedback on our central assumption. We agree that our original wording ("*not* from isolated spikes") was misleading and could imply exclusion.
>
> * Our intention was inclusion. As Reviewer `nbGe` correctly noted, real-world anomalies are diverse, and as Reviewer `u6SV` suggested, concrete scenarios are essential to clarify the model's scope. We have revised the manuscript to state this explicitly.
>
> 1. Theoretical Clarification: A Unified Mechanism
>
> * Our model treats both anomaly types within the same dynamical framework. In our fold-bifurcation ODE (Eq. 7), the risk state $z(t)$ integrates stress $S(t)$:
>   * Gradual Anomaly: A small $S(t)$ accumulating over a long period pushes $z(t)$ toward the threshold.*This models phenomena like gradual component degradation or sensor drift in spacecraft telemetry, which is a key anomaly type in the SMAP/MSL datasets[1]*.
>   * Abrupt Spike: A large-magnitude $S(t)$ over a short duration (impulse) results in a rapid integration $\Delta z \approx \alpha \int S(\tau)d\tau$, causing $z(t)$ to cross the threshold instantaneously. This models **voltage surges** or **sudden sensor glitches**.
>
> * Thus, our framework captures both types within the same dynamical mechanism. We have revised the paper text (as shown in the revision) to explicitly state this.
>
> 2. Empirical Evidence & Concrete Scenarios
>
> * This unified capability is not just theoretical. As shown in `Table 1`, our method achieves SOTA performance on benchmarks dominated by both types of anomalies:
>   * Gradual Failures: Strong performance on SMAP and MSL.
>   * Abrupt Spikes: Top-tier performance on NAB (Abrupt) and UCR (Periodic/Noise).
>
> * To further clarify this, we have added a case study visualization in `the Appendix G (Figure 10, 11)` that directly contrasts our method's $\mathbf{z}(t)$ trajectory on a gradual failure (from SMAP) versus an abrupt spike (from NAB), showing how the model's dynamics naturally adapt to both.

---

> ### Author Response · Authors · 2025-11-21
>
> ## W2. Linkage between Theory and Application
>
> * We appreciate your insightful comment on the need to ground our ODE modeling in specific application domains. We agree that mapping the mathematical formulation to concrete domain variables allows practitioners to better understand the practical value of our method.
>
> * To address this, we adopt the classifications from the TSB-AD benchmark [2]—the source of our datasets. This benchmark categorizes datasets by both application domain and anomaly type (e.g., Seq vs. P&Seq).
>
> 1. **Mapping Formulation to TSB-AD Domain Variables & Constraints**
>
> * We translate the abstract ODE terms ($S, z, Z_{thr}$) into tangible physical concepts for the two primary categories:
>
> | Our method Formulation | Domain A: Physical Systems (TSB-AD: Seq type)<br>(e.g., SMAP, MSL [1], SWaT [3]) | Domain B: IT & Service Systems (TSB-AD: P&Seq type)<br>(e.g., NAB [4], YAHOO [5]) |
> | :--- | :--- | :--- |
> | Stress Input $S(t)$ | Sensor Deviations: Instantaneous fluctuations in telemetry (e.g., voltage, valve pressure) often caused by environmental noise. | Workload Spikes: Sudden increases in CPU load, disk I/O, or network traffic requests. |
> | Risk State $z(t)$ | System Degradation: The accumulated health status reflecting mechanical wear or battery fade (e.g., gradual drift in spacecraft telemetry [1]). | System Instability: The saturation level of server resources. A high $z(t)$ implies the system is nearing a crash or timeout. |
> | Threshold $Z_{thr}$ | Safety/Physical Limit: The operational bounds defined by physics or safety protocols (e.g., "Tank overflow level" in SWaT). | Service Level Agreement (SLA): The maximum tolerable latency or load before service failure occurs. |
>
> 2. **Operational Workflows & Empirical Proof**
>
> * This mapping defines actionable workflows that our model's performance already validates.
>
>     * Scenario 1: Gradual Failure (TSB-AD Seq Category)
>
>       * Context: In datasets like SMAP (Satellite) or SWaT (Water Treatment), anomalies are classified as Seq (Sequence) type, often manifesting as slow drifts due to component degradation.
>       * Our model's SOTA performance on SMAP and MSL (Table 1) *empirically proves* that our ODE mechanism successfully tracks this slow accumulation where other models often fail. This validates the workflow where operators monitor $z(t)$ for **accumulated stress** and trigger preventative maintenance *before* $Z_{thr}$ is breached.
>
>   * Scenario 2: Abrupt Failure (TSB-AD P&Seq Category)
>
>       * Context: In datasets like YAHOO (Web Traffic) or NAB (Real Operational Signals [4]), anomalies are classified as P&Seq, including sharp, isolated spikes.
>       * Our SOTA performance on NAB and UCR (Table 1) confirms that the same ODE mechanism handles these "rapid saturation" events (spikes) effectively. This validates the workflow where the resilience parameter ($\gamma$) filters transient bursts, but triggers an immediate alert when $z(t)$ breaches the SLA threshold ($Z_{thr}$).
>
> * By distinguishing these workflows and backing them with our experimental results, we demonstrate that our method is a single, robust mechanism adaptable to both the "slow accumulation" of Seq anomalies and the "rapid saturation" of P&Seq anomalies.
>
> ## W3: Visualization of Comparative Performance & Uncertainty
>
> To address the concern regarding visual clarity and confidence assessment, we have added two new figures in Appendix F:
>
> 1. **Qualitative Comparison** (`Figure 10 in Appendix G`):
>
> * We visually compare our method against recent SOTA baselines (NRDetector, TranAD) across three distinct anomaly types:
>   * Gradual Drift (SMAP): our method accurately tracks the slow onset of failure, whereas window-based TranAD[6] fails to capture the early accumulation phase.
>   * Abrupt Spike (NAB): our method instantaneously detects the sharp spike, while NRDetector[7] exhibits false positives in noisy regions.
> * This side-by-side comparison demonstrates our method's robustness across diverse failure modes at a glance.
>
> 2. **Uncertainty Assessment** (`Figure 5 & 11 in Section 4 and Appendix G`):
>
> * We visualized the uncertainty bands (derived from MC dropout) to demonstrate model reliability.
>   * Normal State: The bands remain narrow, indicating high confidence.
>   * Anomaly Onset: The bands significantly widen at the moment of anomaly, visually confirming that our method's detection is driven by an intrinsic increase in risk and uncertainty, rather than random fluctuations.

---

> ### Author Response · Authors · 2025-11-21
>
> ## W4: Thresholding Stability & Priors
>
> We deeply appreciate your concern regarding the stability of threshold selection. We address this by (1) explaining the physical necessity of varying thresholds and how priors are incorporated, and (2) verifying the model's robustness using a threshold-independent metric (VUS-PR).
>
> 1. **Why Thresholds Vary & Incorporating Domain Priors**
>
> * The variation in threshold values ($Z_{thr}$) across datasets is not an artifact of instability, but a reflection of the distinct physical scales of different systems.
>   * Physical Scale: For instance, the "accumulated risk" $z(t)$ for a voltage sensor in a satellite (SMAP) operates on a completely different energy scale than the CPU load stress in a server cluster (SMD). Therefore, the absolute values of $Z_{thr}$ must naturally differ to match the system's normal operating range.
>   * Incorporating Priors: We address this by using the distribution of normal data as a data-driven prior to calibrate the baseline.
>
> Furthermore, as you suggested, domain knowledge is explicitly incorporated via the margin parameter $\rho$ in Eq. (9). In practice, domain experts can adjust $\rho$ to inject a specific "safety margin" (e.g., a stricter margin for critical infrastructure), thereby improving calibration stability based on operational requirements.
>
> 2. **Verification via Threshold-Independent Evaluation (VUS-PR)**
>
> * To further alleviate concerns regarding the sensitivity of performance to specific threshold settings, we extended our evaluation to include VUS-PR (Volume Under Surface - Precision Recall).
>   * VUS-PR is a threshold-independent metric recently established as a standard for rigorous anomaly detection [2], as it evaluates performance across all possible decision boundaries rather than a single cut-off.
>   * As shown in the table below (subset of `Table 1 & 2 in Section 4` in the revised paper), our method achieves state-of-the-art performance on VUS-PR as well.
>
> | Setting | Method | Exathlon | MSL | OPPORTUNITY | SMAP | SMD | **Avg Rank** |
> | :--- | :--- | :--- | :--- | :--- | :--- | :--- | :--- |
> | **Univariate** | **OURS** | **0.95** | **0.70** | **0.94** | **0.82** | **0.84** | **1** |
> | | TSPulse (FT) | 0.87 | 0.66 | 0.06 | 0.74 | 0.80 | 2.4 |
> | | TSPulse (ZS) | 0.83 | 0.64 | 0.06 | 0.71 | 0.78 | 3.2 |
> | | Sub-PCA | 0.93 | 0.51 | 0.91 | 0.52 | 0.45 | 3.2 |
> | **Multivariate** | **OURS** | **0.93** | **0.50** | **0.82** | **0.45** | **0.46** | **1** |
> | | TSPulse (FT) | 0.91 | 0.21 | 0.07 | 0.32 | 0.36 | 2.4 |
> | | TSPulse (ZS) | 0.91 | 0.20 | 0.07 | 0.30 | 0.35 | 3.2 |
> | | CNN | 0.68 | 0.35 | 0.16 | 0.19 | 0.35 | 3.2 |
>
> *(Note: This table shows a subset of results. Please refer to the revised manuscript in `Table 1 and 2 in Section 4` for the full 40-dataset benchmark.)*
>
> **Experimental Results:**
> * The fact that our method ranks 1st on average using VUS-PR confirms that our model's performance stems from the inherent separability of the risk score $z(t)$ generated by our ODE dynamics, rather than dependency on a finely-tuned threshold.

---

> ### Author Response · Authors · 2025-11-21
>
> ## W5: Systematic Efficiency Analysis: Asymptotic Complexity & Practical Speed
>
> To address your requests for a deeper analysis of the "sources of improvement" (u6SV) and the "runtime complexity" (nbGe), we provide a systematic breakdown of our method's computational cost.
>
> Our core advantage lies in maintaining Linear $O(L)$ Complexity throughout both training and inference, in sharp contrast to $O(L^2)$ Transformer-based SOTA models.
>
> 1. **Stage 1: Offline Training Cost (The Backbone)**
> * Source of Improvement: The efficiency gains begin here. Many competing SOTA models (e.g., Anomaly Transformer) are built on Transformer backbones with $O(L^2)$ (Quadratic) complexity.
> * our method's Cost: We intentionally use lightweight backbones like DLinear or TimeMixer, which have a Linear $O(L)$ complexity. This makes our initial training phase asymptotically faster.
>
> 2. **Stage 2: Inference Scoring Cost (The Stress Signal)**
> This stage involves calculating the stress $S(t)$ by evaluating the effect of $P$ distinct patches. We also incorporate MC Dropout to estimate the uncertainty term.
>
> * Standard Practice for Reliability: Utilizing MC Dropout for uncertainty quantification is a widely adopted standard practice in trustworthy machine learning. While this introduces a constant multiplier (number of MC samples) to the inference steps, it provides essential reliability without altering the fundamental complexity class.
> * Theoretical Complexity (nbGe): Even with MC Dropout, the total cost is $O(K \cdot P \cdot C_{{pred\_{ours}}})$, where $K$ (MC samples) and $P$ (patches) are small constants independent of $L$. Since our backbone cost $C_{{pred\_{ours}}}$ is $O(L)$, the total scoring complexity strictly maintains **$O(L)$ (Linear)** behavior.
> * Source of Improvement (u6SV): This is the key difference. Our $O(L)$ total inference cost—even with rigorous uncertainty estimation—is **fundamentally (asymptotically) superior** to competitors' $O(L^2)$ inference cost.
> * Real-time Feasibility (nbGe): Furthermore, these computations are massively parallelizable. The independent patch evaluations and MC dropout passes can be processed in a **single batch** on a GPU, keeping the practical wall-clock time extremely fast.
>
> 3. **Stage 3: Inference Integration Cost (Modeling ODE)**
> * our method's Cost: This final step involves the scalar ODE integration. As this operates on the resulting scores, its cost is **negligible** compared to the neural network inference costs in Stage 2.
>
>
> **Why our method is Efficient**
>
> * Theoretically (nbGe): Our total runtime complexity is $O(L)$, which is asymptotically superior to $O(L^2)$ Transformer-based detectors.
> * Empirically (u6SV): The "source of improvement" is our Linear-by-design backbone. This is empirically validated by the minimal GPU/system memory usage reported in Table 11.
> * Practically (nbGe): **Our our method's anomaly detector is a training-free extension of the forecasting model. It is derived directly from the pre-trained forecaster without additional optimization. Combined with highly parallelizable inference computations, this makes our method ideal for real-time deployment.**
>
>
> ---
> [1] Hundman, K., Constantinou, V., Laporte, C., Colwell, I., & Soderstrom, T. (2018). Detecting Spacecraft Anomalies Using LSTMs and Nonparametric Dynamic Thresholding. *Proceedings of the 24th ACM SIGKDD International Conference on Knowledge Discovery & Data Mining*.
>
> [2] Liu, Q., & Paparrizos, J. (2024). The Elephant in the Room: Towards A Reliable Time-Series Anomaly Detection Benchmark. Advances in Neural Information Processing Systems, 37.
>
> [3] Mathur, A. P., et al. "SWaT: A water treatment testbed..." CySWater 2016.
>
> [4] Lavin, A., et al. "Evaluating Real-Time Anomaly Detection..." ICMLA 2015. (Source of NAB)
>
> [5] Laptev, N., et al. "S5 - A Labeled Anomaly Detection Dataset..." 2015. (Source of Yahoo)
>
> [6] Shreshth Tuli, Giuliano Casale, and Nicholas R Jennings. Tranad: Deep transformer networks for
> anomaly detection in multivariate time series data. Proceedings of the VLDB Endowment, 15(6):
> 1201–1214, 2022
>
> [7] Yaxuan Wang, Hao Cheng, Jing Xiong, Qingsong Wen, Han Jia, Ruixuan Song, Liyuan Zhang,
> Zhaowei Zhu, and Yang Liu. Noise-resilient point-wise anomaly detection in time series using
> weak segment labels. arXiv preprint arXiv:2501.11959, 2025.

---

> ### Author Response · Authors · 2025-11-27
>
> Dear Reviewer u6SV,
>
> We sincerely thank you for your constructive guidance. As the deadline approaches, we would be very grateful for your thoughts on our revisions.
>
> To address your concerns on "Theory to Practice," we have added concrete domain mappings (Physical vs. IT) and operational workflows. We also updated the visualizations with uncertainty bands and clear comparative plots against SOTA baselines to demonstrate our method's reliability and performance advantage "at a glance."
>
> We have endeavored to address every point you raised with the utmost care. We hope these updates meet your expectations and warrant a stronger recommendation.

---

> ### Comment · Reviewer_u6SV · 2025-11-27
>
> I thank the authors for their response. The claim that the model treats both anomaly types within the same dynamical framework is still not clearly justified. It is not sufficiently explained, in a formal and precise way, how both anomaly types are modeled within a single framework, what assumptions enable this unification, and whether they truly arise from one underlying model rather than from ad-hoc design choices. The paper also repeatedly claims that the method is robust and consistent across settings, but there is no theoretical analysis or deeper explanation to support this.

---

> ### Author Response · Authors · 2025-11-27
> **Formal Justification: The Unified Mechanism is Intrinsic, Not Ad-hoc**
>
> We respectfully clarify that our unified handling of anomalies is not an ad-hoc design choice, but a fundamental mathematical consequence of the proposed ODE. Our inference process is strictly invariant: we calculate $S(t)$ and solve the same fixed ODE (Eq. 7) regardless of the data type.
> We provide the formal derivation showing how this single equation naturally covers both scenarios and ensures robustness.
>
> 1. One Equation, Two Regimes (Mathematical Derivation)
>
> The system state $z(t)$ evolves according to:
>
> $\frac{dz}{dt} = \alpha S(t) - (\beta z^2 + \gamma z)$, where $\alpha S(t)$ refers to **forcing** and $(\beta z^2 + \gamma z)$ refers to **restoring**.
>
> We derive how the dominant terms in this equation automatically shift based on the input timescale of $S(t)$:
> * Case A: Gradual Accumulation (Quasi-static Regime)
>     * Condition: $S(t)$ varies slowly ($\frac{dS}{dt} \approx 0$).
>     * Dynamics: The time derivative $\dot{z}$ is negligible. The system tracks the moving equilibrium where Forces Balance: $\alpha S \approx \beta z^2 + \gamma z$.
>     * Result: $z(t)$ gradually tracks the rising stress. Anomaly detection occurs via Bifurcation (loss of stability) when stress exceeds the system's capacity.
> * Case B: Abrupt Spike (Impulsive Regime)
>     * Condition: $S(t)$ is a large magnitude impulse over a short duration $\Delta t$ ($S \gg \beta z^2 + \gamma z$).
>     * Dynamics: The forcing term dominates the restoring terms. The ODE asymptotically simplifies to an Integrator: $\frac{dz}{dt} \approx \alpha S$.
>     * Result: $z(t)$ jumps instantaneously by the integrated stress energy ($\Delta z \approx \int \alpha S dt$), crossing the threshold immediately.
>
> * $\therefore$  Conclusion: The model does not know the anomaly type. The transition between "tracking equilibrium" and "integrating impulses" is a mathematical necessity of the ODE dynamics.
>
>
> 2. Theoretical Proof of Robustness (Restoration Mechanism)
> You questioned the theoretical basis for robustness. This is guaranteed by the Linear Stability Analysis of the restoration term ($-\gamma z$).
> * Stability Proof: Near the stable equilibrium, the Jacobian eigenvalue is $\lambda \approx -\gamma < 0$. This guarantees that any perturbation (noise) decays exponentially ($e^{-\gamma t}$).
> * Implication: The system is theoretically forced to return to the normal state once the stress subsides. This confirms that robustness is structurally enforced by the equation, not by ad-hoc filtering.
>
>
> 3. Visual Confirmation (Evidence of Restoration)
> * Sudden Spike: The green line in Figure 12 (c) and Figure 13 (a) (top panel) shows a sudden spike in the input features.
> * Instant Reaction: Correspondingly, our risk state $z(t)$ (blue line, bottom panel) reacts instantly, shooting up and crossing the threshold (red dashed line), correctly flagging the anomaly.
> * Restoration: Crucially, immediately after the spike, the risk state $z(t)$ rapidly decays back to the stable basin (low values) due to the restoring term ($-\gamma z$). Since there is no further accumulated stress, the state remains stable afterwards.
> This visual evidence perfectly matches our theoretical derivation: the model is sensitive enough to catch the spike (Impulsive Regime) yet robust enough to recover immediately (Restoration Mechanism) without manual intervention.

---

> ### Author Response · Authors · 2025-11-28
>
> To ensure our previous response was fully clear, we would like to provide a step-by-step walkthrough of how the single Fold-bifurcation ODE mechanism processes different anomaly types without any manual intervention.
> 1. The Invariant Mechanism
> Our inference process is identical for every time step $t$:
>
> * 1. Input: Receive stress $S(t)$ from the forecaster. (In Section 3.2)
> * 2. Process: Update state $z(t)$ using the fixed equation: $\frac{dz}{dt} = \alpha S(t) - \beta z(t)^2 - \gamma z(t)$.
> * 3. Output: If $z(t) > Z_{thr}$, flag anomaly.
>
>
> 2. Mathematical Derivation: Time-Scale Separation Analysis
> The unified capability of the model is a direct consequence of Time-Scale Separation. We analyze the behavior of the governing ODE (Eq. 7) by comparing the characteristic time-scale of the input stress, $\tau_S$, with the system's intrinsic relaxation time, $\tau_{sys} \approx 1/\gamma$.
> * Scenario A: Quasi-Static Regime (Gradual Drift)
>     * Condition: The stress $S(t)$ varies slowly relative to the system dynamics ($\tau_S \gg \tau_{sys}$), implying that the time derivative term $|\frac{dz}{dt}|$ is negligible compared to the algebraic terms.
>     * Formal Approximation: In this limit, the system operates on a slow manifold, maintaining an instantaneous force balance where the external forcing equals the restoring dynamics:
>     * $$\frac{dz}{dt} \approx 0 \implies \alpha S(t) \approx \beta z(t)^2 + \gamma z(t)$$
>     * Mechanism (Adiabatic Tracking): Consequently, the risk state $z(t)$ is algebraically constrained to adiabatically track the magnitude of $S(t)$. Anomaly detection is triggered when $S(t)$ grows sufficiently to destabilize this equilibrium (i.e., crossing the bifurcation point).
> * Scenario B Regime 2: Impulsive Limit (Abrupt Spikes)
>     * Condition: The stress $S(t)$ acts as a large-magnitude impulse over a short duration $\Delta t$ within the full sequence.
>     * Global Integration: Although we integrate the ODE over the entire sequence ($t=1 \dots L$), during this specific short interval $\Delta t$, the forcing term $\alpha S(t)$ dominates the restoring terms ($\alpha S \gg \beta z^2 + \gamma z$).
>     * Local Behavior: Consequently, in this window, the ODE locally acts as an Integrator: $\frac{dz}{dt} \approx \alpha S(t)$. This causes $z(t)$ to undergo an instantaneous jump proportional to the impulse energy, triggering the detection.
>
>
> 3. Theoretical Guarantee of Self-Restoration (Robustness)
>
> * As visually confirmed in the SMD example (`Figure 11 c`), the model exhibits a built-in healing mechanism. We formalize this property as Asymptotic Stability.
>
>   * Condition: Immediately after a transient spike subsides, the external forcing vanishes ($S(t) \to 0$). The governing dynamics simplify to the autonomous decay equation:  $$\frac{dz}{dt} = -(\beta z^2 + \gamma z)$$
>
>   * Proof: Since parameters $\beta, \gamma > 0$, the time derivative is strictly negative for any risk state $z(t) > 0$ (note: risk is non-negative by design), enforcing monotonic decay. Furthermore, as $z(t)$ approaches the stable basin (small $z$), the linear damping term dominates the quadratic term ($\gamma z \gg \beta z^2$), leading to exponential relaxation:  $$\frac{dz}{dt} \approx -\gamma z \implies z(t) \propto e^{-\gamma t}$$
>
>   * Conclusion: This proves that the system is mathematically guaranteed to recover to the normal equilibrium state in the absence of sustained stress. This intrinsic stability ensures that the model automatically "resets" itself after shocks, preventing error propagation and false positives in subsequent steps.
>
>
> * We will explicitly integrate this formal mathematical derivation and stability analysis into the revised manuscript in `Appendix H`. We plan to upload the updated version within the next few days to ensure the theoretical foundation is rigorous and self-contained for all readers.
>
> We believe this formal proof, combined with the visual evidence, clearly justifies that the unified capability is an intrinsic property of the ODE, **not an ad-hoc choice**.
> We hope this detailed response resolves your concerns regarding the theoretical justification and allows you to reconsider your assessment once the revision is available. We remain fully available to address any further questions you may have.

---

### Official Review · Reviewer_DxMr · 2025-10-31

**Soundness:** 3
**Presentation:** 3
**Contribution:** 3
**Rating:** 6
**Confidence:** 2

**Summary:**

This paper introduces a point-wise anomaly detection method grounded in fold-bifurcation dynamics. The approach is fully unsupervised, requiring neither anomaly labels nor detector training.

**Strengths:**

The paper develops an intersting view of anomaly detection where by anomaly is seen as the cause of accumulating stress within the system.  Stress signals are extracted from time series data and integrated using a fold-bifurcation ODE.  These signals accumulate over time till they reach a tipping point and an anomalous condition is raised.

The proposed method is reported to address a key shortcoming of prior approaches that show strong results on window-based metrics but often fail to generalize under point-wise evaluations.

The proposed scheme is evaluated on 9 benchmarks and compared against 10 baselines, and the results suggest that the proposed scheme achieves good performance under strict point-wise evaluation.

The premise of this work appears sound given the work in (Scheffer et al. 2009) that shows that variance increases as a system approaches a critical transition.

**Weaknesses:**

Please state the paper’s novelty relative to the early-warning systems you cite---what does this scheme enable that prior EWS methods do not?

Perhaps I missed it, but it seems that z(t) is computed independently for each feature?  Is that so?  If it is true then how does this method captures subtle interactions between multiple variables present in the time series?  I couldn't following Figure 3, for example, since I was confused to see a single z(t) curve.  Shouldn't there be one for each variable?

**Questions:**

- Can you please discuss control parameter r in Eq. 1 and Eq. 2 and why is it appropriate to capture it as a time-varying stress signal S(t) in Eq. 3.

- In what sense is FREE better than early warning systems that monitor Eigenvalues trends?  Why cannot we use early warning systems for anomaly detection?

- Perhaps it is obvious to those working with these ODEs, it isn't immediately obvious to me how will one determine that risk trajectory z(t) has left its stable basin?

- In Sec 3.2, the forecasting model f_{\theta} predicts the next H elements?

- Purturbed sequence contains X except those that belong to patch i?  Is this is what we mean by purturbed sequence?

---

> ### Author Response · Authors · 2025-11-21
>
> We thank Reviewer DxMr for the constructive feedback and the encouraging comments. We appreciate the opportunity to clarify the key mechanisms of our method, particularly regarding the stable basin determination and multivariate handling.
>
> Below, we provide detailed responses to your questions.
>
> ## W1(Q2): Our method vs. Early-Warning System (EWS)
>
> * Eigenvalue-trend EWS provides trend-level warnings (e.g., largest eigenvalue drift) that are informative for impending regime shifts but not tailored to point-wise anomaly decisions: they require long windows, detrending, and yield latency-sensitive, coarse signals—often missing abrupt spikes and offering limited localization in multivariate settings.
>
> * Our method instead produces a point-wise risk trajectory $z(t)$ driven by a data-derived stress $S(t)$, enabling exact time-point decisions (and segments via consecutive exceedances), handling both gradual and abrupt events under the same dynamics, and providing per-feature attribution via $S_k(t)$, where $k$ is a feature dimension.
>
> * Practically, our method's inference is low-latency (one base forecast + small local updates) and thus better suited for real-time industrial/operational anomaly detection, while EWS remains complementary for macro early-warning.
>
> * How our method differs from eigenvalue-based EWS. EWS tracks trend statistics of stability (e.g., eigenvalue drift) that need long windows and yield coarse, lagged signals. our method instantiates a risk dynamics driven by a data-derived stress $S(t)$, producing point-wise decisions (and segments via grouping) with per-feature attribution and low latency.
>
> * In multivariate, our method avoids high-variance covariance/eigenvalue estimation and provides localized explanations through $S_k(t)$ and $z_k(t)$.
>
> ## W2: Multivariate setting & interactions
>
> We clarify how our method captures multivariate interactions and how the risk trajectory $z(t)$ is formulated for evaluation.
>
> 1. Capturing Interactions via the Forecaster ($f_\theta$)
>
> * While the ODE dynamics are computed feature-wise, multivariate interactions are explicitly encoded by the forecasting backbone ($f_\theta$).
>
>   * **Mechanism:** Our backbone models (e.g., PatchTST, TimeMixer) process multivariate inputs and learn cross-channel dependencies.
>   * **Stress Signal:** Consequently, the stress signal $S(t)$ derived from the forecaster captures how a perturbation in one variable affects the entire system. Thus, the inputs to the ODE already reflect multivariate contexts.
>
> 2. Feature-wise Dynamics to System-level Risk
>
> * As described in `Equation 7 in Section 3.3`, we evolve the risk state feature-wise ($\mathbf{z}(t) \in \mathbb{R}^d$) to track the stress accumulation of each variable. However, for the **final anomaly decision**, we aggregate these into a **System-level Risk Score**.
>   * **Aggregation Strategy:** The single curve shown in `Figure 3 in Section 4` and used for detection is the aggregated risk (e.g., taking the mean or max across dimensions: $z_{sys}(t) = \text{Agg}(\mathbf{z}(t))$).
>   * This approach allows the model to detect system-wide failures whether they are caused by a single critical sensor (captured by Max) or a collective degradation (captured by Mean), providing a robust **single indicator** for the operator.
>
> * We have updated `Section 3.3` to explicitly state that the feature-wise risk trajectories are aggregated to form a unified anomaly indicator.
>
> * **Direct answer to R-DxMr.**
>   * Interactions are inherently captured by the multivariate forecaster ($f_\theta$) and reflected in the stress signals. The single curve in `Figure 3 in Section 4` represents the aggregated system-level risk which serves as the actual metric for the final detection decision, rather than just a simplification for plotting. While the detection relies on this system-level score (visualized in `Figure 12 in Appendix G` for multivariate datasets).
>
> ## Q1-(1).  Definition of $r$ as an Exogenous Parameter in Fold Bifurcation ODE in Eq. (1)- (2)
>
> * In the canonical fold-bifurcation ODE ($\frac{d\mathbf{z}(t)}{dt} = \mathbf{r} - \mathbf{z}^2$) (and its damped variant ($\frac{d\mathbf{z}(t)}{dt} = \mathbf{r} - \mathbf{z}^2 - \gamma \mathbf{z})$), $\mathbf{r}$ is an exogenous control/unfolding parameter:
>   * it shifts the equilibria (i.e., the stable and unstable fixed points $\mathbf{z}^\star=\pm\sqrt{\mathbf{r}}$) and
>   * governs their saddle–node (fold) collision at the tipping point [1,2].
>
> * Many studies interpret $\mathbf{r}$ as a slowly varying environmental pressure and analyze the associated loss of resilience prior to tipping [3,4].
> * The linear term ($-\gamma \mathbf{z}$) models resilience/decay and keeps the dynamics dissipative without changing the fold geometry.

---

> ### Author Response · Authors · 2025-11-21
>
> ## Q1-(2).  Justification for Modeling Stress signal $S(t)$ as Control Parameter $r$
>
> * In our method we instantiate the control as a data-driven, time-varying input by setting ($\mathbf{r}(t)\equiv \alpha,\mathbf{S}(t)$), where ($\mathbf{S}(t)$) is a multivariate stress extracted from a forecaster (sensitivity + uncertainty).
>
> * This is appropriate because:
>   1. conceptually, higher stress corresponds to stronger external pressure pushing the state toward the threshold—the same role played by ($\mathbf{r}$) [3]
>   2. mathematically, for slowly varying inputs the instantaneous equilibria ($\beta \mathbf{z}^2+\gamma \mathbf{z}=\alpha \mathbf{S}(t)$) preserve the fold geometry of the autonomous normal form (a standard non-autonomous reduction) [2,5] and
>   3. practically, ($\mathbf{S}(t)$) is multivariate and time-varying, so it captures cross-feature interactions and abrupt shocks observed in data. We will add a small overlay figure showing how increasing ($S(t)$) moves the equilibria and triggers the point-wise crossing.
>
> * Furthermore, to clarify this conceptual link for reviewer (and also readers), we have revised the text in `Section 3.3 (as highlighted in the revision)` to explicitly describe $\mathbf{S}(t)$ as a dynamic external force that drives the system state toward instability, analogous to the control parameter $\mathbf{r}$.
>
> ## Q3. Clarification on Stable Basin Determination
>
> * We fully agree that while the concept of a 'stable basin' is standard in dynamical systems theory, translating it into a concrete decision rule for anomaly detection is a critical practical step.
>
> * We clarify this process through both physical intuition and our data-driven implementation:
>
>   1. **Physical Intuition: The "Tipping Point"**
>   * Conceptually, the "stable basin" can be visualized as a potential well (a bowl) containing a ball (the system state $\mathbf{z}(t)$).
>   * **Normal State:** As long as the external stress $\mathbf{S}(t)$ is manageable, the ball oscillates within the bowl. Even if it moves up the sides, the system's resilience (the $-\gamma \mathbf{z}(t)$ term in Eq. 7) pulls it back toward the equilibrium.
>   * **Leaving the Basin:** If the accumulated stress becomes too large, the ball is pushed over the edge of the bowl.
>   * Mathematically, this corresponds to the "tipping point" or bifurcation point where the stable equilibrium disappears. Once this boundary is crossed, the value of $\mathbf{z}(t)$ increases rapidly (escalates) due to the nonlinear term $-\beta \mathbf{z}(t)^2$.
>
> 2. **Practical Determination: Data-Driven Thresholding ($Z_{thr}$)**
>
> * Instead of solving for the theoretical bifurcation point analytically (which is complex under time-varying inputs), we determine the boundary of the stable basin **empirically** using the normal training data (as detailed in `Section 3.4`):
>
>   * **Calibration:** We simulate the ODE on the normal training data to observe the maximum extent of $\mathbf{z}(t)$ under normal conditions. This defines the "observed" stable basin.
>   * **Thresholding (`Eq. 8 & 9 in Section 3.4` ):** We set the anomaly threshold $Z_{thr}$ as a high percentile (e.g., 99%) of these maximum risk values observed during training.
>     * **Decision Rule:** If $\mathbf{z}(t) > Z_{thr}$ at test time, we determine that the trajectory has "left the stable basin" and flag an anomaly.
>
> 3. **Visual Evidence**
>
> * This mechanism is visually demonstrated in `Figure 3 in Section 4` of our paper.
>   * Top panels (Normal): The risk trajectory $z(t)$ (green line) fluctuates but remains bounded within the calibrated threshold (red dashed line).
>   * Bottom panels (Test): When an anomaly occurs, the stress accumulates, causing $z(t)$ to abruptly shoot up and cross the $Z_{thr}$ line. This crossing point effectively marks the moment the system leaves its stable basin.
>
> * To make this link explicit for readers, we have revised `Section 3.4` to state that
>   > "we interpret the threshold $Z_{thr}$ as the empirical boundary of the stable basin,"
>
>     clarifying that exceeding this threshold is the practical definition of leaving the stable state.
>
> ## Q4. the forecasting model $f_{\theta}$ predicts the next $H$ elements?
>
> * Yes, that is correct. As described in `Section 3.2`, the forecasting model $f_{\theta}$ takes the input sequence $X$ (or the masked sequence $X_{\setminus P_i}$) and predicts the subsequent $H$ time steps (Forecasting Horizon). This prediction is then used to compute the sensitivity and uncertainty terms for the stress signal $S(t)$.

---

> ### Author Response · Authors · 2025-11-21
>
> ## Q5. Definition of purturbed sequence
>
> * Yes. By perturbed sequence ($X_{\setminus i}$), we mean the original input sequence $X$ where the specific patch $P_i$ has been masked out (replaced with zero or a mask token). This allows us to measure the model's sensitivity to the missing information in that specific patch.
>
> * To prevent ambiguity regarding whether the patch is removed or masked, we have clarified the definition in `Section 3.2` to explicitly state that the segment is replaced with a mask token.
>
> ---
> [1] Strogatz, *Nonlinear Dynamics and Chaos*, 2nd ed., 2015.
>
> [2] Kuehn, “A mathematical framework for critical transitions: A survey,” *Physica D*, 2011.
>
> [3] Scheffer et al., “Early-warning signals for critical transitions,” *Nature*, 2009.
>
> [4] Lenton et al., “Tipping elements in the Earth’s climate system,” *PNAS*, 2008.
>
> [5] Ashwin, Wieczorek, Vitolo, Cox, “Tipping points in open systems: bifurcation, noise-induced and rate-dependent phenomena,” *Phil. Trans. R. Soc. A*, 2012.

---

> ### Author Response · Authors · 2025-11-27
>
> Dear Reviewer DxMr,
>
> We deeply appreciate your time and valuable insights. We are writing to kindly request your feedback on our rebuttal before the discussion period ends. We have provided detailed clarifications on the novelty against EWS and the multivariate handling mechanism (via the forecaster backbone). We also revised the manuscript to make the theoretical connections (e.g., parameter $r$ vs. $S(t)$) more explicit based on your questions.
>
> We hope our responses have resolved your queries, and we would greatly appreciate your re-evaluation.

---

### Official Review · Reviewer_nbGe · 2025-10-31

**Soundness:** 3
**Presentation:** 3
**Contribution:** 2
**Rating:** 2
**Confidence:** 4

**Summary:**

The paper proposed a method for point anomaly detection in time series, which is an important task in time series domain. The proposed method employs forecasting model and provide masked signal to produce a sensitivity score, which is then utilized with the fold bifurcation theory to get anomaly scores. The method seems to outperform some baselines across 9 datasets in F1 score.

**Strengths:**

The paper has the following strong points:
1. The proposed method is novel, and well motivated. Particularly, the paper does a good job in motivating the need for the fold bifurcation theory and it's usage for anomaly detection.
2. The proposed method is theoretically motivated and well formulated.
3. Some experimental evidence has been provided.
4. The paper is well written.

**Weaknesses:**

However, the paper has the following weaknesses.
1. Does the proposed method work only for point anomalies? The author(s) say "However, most approaches are evaluated under coarse window-level settings, which can mask their limitations in the stricter point-wise anomaly detection scenario." This is not entirely true. There are recent papers with multiple types of anomaly detection capabilities. Example include: [1], [2], and many more.
2. The evaluation metric F1 score has been establised as a biased metric in recent literature [1]. Why does the author not use mathematically proven good metrics like VUS-PR defined in [1]?
3. The evaluation should be done on well establised anomaly detection benchamarks such as in [1], and should be compared with recent SOTA methods on more datasets.
4. Since the proposed method does patching, masking etc., what is the runtime complexity of the method? In terms for both theoretical and empirical evidence. Since anomaly detection is needed in realtime in most usecases, runtime is a very important factor.
5. The authors say "many real-world failures arise not from isolated spikes but from the gradual accumulation of stress that drives a system toward a critical transition." This is not true in several situation when external factors create anomalies, or even periodic anomalies.
6. How does the method perform on multivariate time series?

[1] Liu, Qinghua, and John Paparrizos. "The elephant in the room: Towards a reliable time-series anomaly detection benchmark." Advances in Neural Information Processing Systems 37 (2024): 108231-108261.
[2] Ekambaram, V., Kumar, S., Jati, A., Mukherjee, S., Sakai, T., Dayama, P., ... & Kalagnanam, J. (2025). TSPulse: Dual Space Tiny Pre-Trained Models for Rapid Time-Series Analysis. arXiv preprint arXiv:2505.13033.

**Questions:**

See weaknesses.

---

> ### Author Response · Authors · 2025-11-21
>
> We sincerely thank Reviewer nbGe for the sharp and highly constructive feedback. In particular, we appreciate your crucial correction regarding our claims on existing literature and your strong recommendation to adopt the TSB-AD benchmark and VUS-PR metric. Taking your advice to heart, we have significantly expanded our experimental scope and revised our manuscript to ensure a fair and rigorous evaluation.
>
> ## W1: Point vs. Range Anomalies and Wording Clarification
>
> 1. **Wording Clarification (Response to "Not entirely true")**
>
> * We acknowledge the reviewer's point. Our original statement was intended to highlight that strict point-wise evaluation has been relatively scarce compared to the long-standing dominance of window-level protocols in the broader literature. However, we agree that this phrasing could be misinterpreted as overlooking the emerging wave of recent rigorous studies (e.g., [2], [3]).
>
> * To prevent such misunderstanding and to accurately reflect the current SOTA landscape, we have revised the Abstract:
>
>   > “While many prior works report window-level metrics that may mask errors, several recent methods evaluate at the point level as well. Our goal is to use a stricter point-wise protocol to make masking effects explicit.”
>
> 2. **Scope of Detection (Response to "Only for point anomalies?")**
>
> * Our method is **not** limited to point anomalies. While we evaluate point-wise to avoid masking, our method naturally handles **segment/collective anomalies**:
>   * We compute risk $z(t)$ at every time step.
>   * A **range anomaly** is simply detected as a contiguous run of point-wise exceedances ($z(t) > Z_{thr}$).
>   * Thus, we do not exclude range anomalies; rather, we evaluate their detection precision at the finest granularity.
>
> 3. **Baselines**
>
> * We already compare against **NRDetector** (a recent point-wise method). Furthermore, as detailed in **W2**, we have explicitly added 34 baselines (including TSPulse [2]) to our benchmarks to ensure a fair comparison with the latest point-wise capabilities.
>
>
> ## W2, W3: Additional Baselines, Threshold-Independent Metrics, and Ranking Analysis
>
> To directly address the reviewer’s concern regarding evaluation bias and benchmark coverage, we have expanded our experiments significantly by incorporating TSPulse [2] and extending our comparison to include **34 baselines** from the **TSB-AD benchmark** [3].
>
> 1. **Extensive Benchmarking & VUS-PR Adoption**
>
> * We evaluated all methods using **VUS-PR (Volume Under Surface - Precision Recall)**, as recommended by [3], to eliminate threshold selection bias.
> * **Result:** our method maintains SOTA performance under VUS-PR. To demonstrate this, we present a summary of VUS-PR scores on representative datasets from the TSB-AD benchmark below (full results in  `Tables 1 & 2 in Section 4.1` of the revised paper).
>
> | Setting | Method | Exathlon | MSL | OPPORTUNITY | SMAP | SMD | **Avg. Rank** |
> | :--- | :--- | :--- | :--- | :--- | :--- | :--- | :--- |
> | **Univariate** | **OURS** | **0.95** | **0.70** | **0.94** | **0.82** | **0.84** | **1** |
> | | TSPulse (FT) | 0.87 | 0.66 | 0.06 | 0.74 | 0.80 | 2.4 |
> | | TSPulse (ZS) | 0.83 | 0.64 | 0.06 | 0.71 | 0.78 | 3.2 |
> | | Sub-PCA | 0.93 | 0.51 | 0.91 | 0.52 | 0.45 | 3.2 |
> | **Multivariate** | **OURS** | **0.93** | **0.50** | **0.82** | **0.45** | **0.46** | **1** |
> | | TSPulse (FT) | 0.91 | 0.21 | 0.07 | 0.32 | 0.36 | 2.4 |
> | | TSPulse (ZS) | 0.91 | 0.20 | 0.07 | 0.30 | 0.35 | 3.2 |
> | | CNN | 0.68 | 0.35 | 0.16 | 0.19 | 0.35 | 3.2 |
>
> *(Note: This table shows a subset of results. Please refer to the revised manuscript for the full 40-dataset benchmark.)*
>
> 2. **Global Ranking Analysis**
>
> * We conducted a comprehensive ranking analysis across the entire TSB-AD dataset suite.
>
>   * **Comparative Scope:** We compared our method against 43 baselines, including recent SOTA models like TSPulse, MOMENT, TranAD and TimesNet.
>   * **Conclusion:** As shown in the Avg. Rank column of the full tables (`Table 1 & 2 in Section 4.1` in the revised paper), **our method achieves the best (lowest) average rank** across both Univariate and Multivariate settings. This confirms that while specific statistical baselines may excel on individual simple datasets, our method offers the most robust and consistent performance across diverse domains.
>
> 3. **Reproducibility**
> * To ensure transparency and allow for verification of these additional results, we have updated the anonymous code (linked in the paper) with scripts and configurations used for the TSB-AD benchmark and VUS-PR evaluation.

---

> ### Author Response · Authors · 2025-11-21
>
> ## W4: Systematic Efficiency Analysis: Asymptotic Complexity & Practical Speed
>
> We appreciate the constructive feedback regarding our model's efficiency. To address your requests for a deeper analysis of the "sources of improvement" (u6SV) and the "runtime complexity" (nbGe), we provide a systematic breakdown of our method's computational cost.
>
> Our core advantage lies in maintaining Linear $O(L)$ Complexity throughout both training and inference, in sharp contrast to $O(L^2)$ Transformer-based SOTA models.
>
> 1. **Stage 1: Offline Training Cost (The Backbone forecasting model)**
>
> * Source of Improvement: The efficiency gains begin here. Many competing SOTA models (e.g., Anomaly Transformer) are built on Transformer backbones with $O(L^2)$ (Quadratic) complexity.
> * Our method's Cost: We intentionally use lightweight backbones like DLinear or TimeMixer, which have a **Linear $O(L)$ complexity**. This makes our initial training phase asymptotically faster.
>
> 2. **Stage 2: Inference Scoring Cost (The Stress Signal)**
>
> * This stage involves calculating the stress $S(t)$ by evaluating the effect of $P$ distinct patches. We also incorporate MC Dropout to estimate the uncertainty term.
>
>   * Standard Practice for Reliability: Utilizing MC Dropout for uncertainty quantification is a **widely adopted standard practice** in trustworthy machine learning. While this introduces a constant multiplier (number of MC samples) to the inference steps, it provides essential reliability without altering the fundamental complexity class.
>   * Theoretical Complexity (nbGe): Even with MC Dropout, the total cost is $O(K \cdot P \cdot C_{pred\_{ours}})$, where $K$ (MC samples) and $P$ (patches) are small constants independent of $L$. Since our backbone cost $C_{pred\_{ours}}$ is $O(L)$, the total scoring complexity strictly maintains $O(L)$ (Linear) behavior.
>   * Source of Improvement (u6SV): This is the key difference. Our $O(L)$ total inference cost—even with rigorous uncertainty estimation—is fundamentally (asymptotically) superior to competitors' $O(L^2)$ inference cost.
>   * Real-time Feasibility (nbGe): Furthermore, these computations are massively parallelizable. The independent patch evaluations and MC dropout passes can be processed in a single batch on a GPU, keeping the practical wall-clock time extremely fast.
>
> 3. **Stage 3: Inference Integration Cost (The Fold-bifurcation ODE)**
>
> * Our method's Cost: This final step involves the scalar ODE integration. As this operates on the resulting scores, its cost is **negligible** compared to the neural network inference costs in Stage 2.
>
> ## W5: Clarification of the "Stress Accumulation" Assumption
>
> We thank both reviewers(nbGe and u6SV) for their valuable feedback on our central assumption. We agree that our original wording ("*not* from isolated spikes") was misleading and could imply exclusion.
>
> Our intention was inclusion. As Reviewer `nbGe` correctly noted, real-world anomalies are diverse, and as Reviewer `u6SV` suggested, concrete scenarios are essential to clarify the model's scope. We have revised the manuscript to state this explicitly in `Appendix H`.
>
> 1. **Theoretical Clarification: A Unified Mechanism**
>
> * Our model treats both anomaly types within the same dynamical framework. In our fold-bifurcation ODE (`Eq. 7 in Section 3.3`), the risk state $z(t)$ integrates stress $S(t)$:
>   * Gradual Anomaly: A small $S(t)$ accumulating over a long period pushes $z(t)$ toward the threshold.*This models phenomena like gradual component degradation or sensor drift in spacecraft telemetry, which is a key anomaly type in the SMAP/MSL datasets[1]*.
>   * Abrupt Spike: A large-magnitude $S(t)$ over a short duration (impulse) results in a rapid integration $\Delta z \approx \alpha \int S(\tau)d\tau$, causing $z(t)$ to cross the threshold instantaneously. This models voltage surges or sudden sensor glitches.
>
> * Thus, our framework captures both types within the same dynamical mechanism. We have revised the paper text (as shown in the revision `Appendix H`) to explicitly state this.
>
> 2. **Empirical Evidence & Concrete Scenarios**
>
> * This unified capability is not just theoretical. As shown in `Table 1 in Section 4.1`, our method achieves SOTA performance on benchmarks dominated by both types of anomalies:
>   * Gradual Failures: Strong performance on SMAP and MSL.
>   * Abrupt Spikes: Top-tier performance on NAB (Abrupt) and UCR (Periodic/Noise).
>
> * To further clarify this, we have added a case study visualization in the `Figure 10,11 in Appendix G` that directly contrasts our method's $z(t)$ trajectory on a gradual failure (from SMAP) versus an abrupt spike (from NAB), showing how the model's dynamics naturally adapt to both.

---

> ### Author Response · Authors · 2025-11-21
>
> ## W6: Multivariate setting & interactions
>
> We clarify how our method captures multivariate interactions and how the risk trajectory $z(t)$ is formulated for evaluation.
>
> 1. **Capturing Interactions via the Forecaster ($f_\theta$)**
>
> * While the ODE dynamics are computed feature-wise, multivariate interactions are explicitly encoded by the forecasting backbone ($f_\theta$).
>   * Mechanism: Our backbone models (e.g., DLinear, PatchTST, TimeMixer) process multivariate inputs and learn cross-channel dependencies.
>   * Stress Signal: Consequently, the stress signal $S(t)$ derived from the forecaster captures how a perturbation in one variable affects the entire system. Thus, the inputs to the ODE already reflect multivariate contexts.
>
> 2. **Feature-wise Dynamics to System-level Risk**
>
> * As described in `Eq. 7 in Section 3.3 in Section 3`, we evolve the risk state feature-wise ($\mathbf{z}(t) \in \mathbb{R}^d$) to track the stress accumulation of each variable. However, for the final anomaly decision, we aggregate these into a System-level Risk Score.
>   * Aggregation Strategy: The single curve shown in `Figure 3 in Section 3` and used for detection is the aggregated risk (e.g., taking the mean or max across dimensions: $\mathbf{z}_{sys}(t) = \text{Agg}(\mathbf{z}(t))$).
>   * This approach allows the model to detect system-wide failures whether they are caused by a single critical sensor (captured by Max) or a collective degradation (captured by Mean), providing a robust single indicator for the operator.
>
> We have updated `Section 3.3` and `Appendix B.4` to explicitly state that the feature-wise risk trajectories are aggregated to form a unified anomaly indicator.
>
> ---
> [1] Hundman, K., Constantinou, V., Laporte, C., Colwell, I., & Soderstrom, T. (2018). Detecting Spacecraft Anomalies Using LSTMs and Nonparametric Dynamic Thresholding. *Proceedings of the 24th ACM SIGKDD International Conference on Knowledge Discovery & Data Mining*.
>
> [2] Ekambaram, V., Kumar, S., Jati, A., Mukherjee, S., Sakai, T., Dayama, P., ... & Kalagnanam, J. (2025). TSPulse: Dual Space Tiny Pre-Trained Models for Rapid Time-Series Analysis. arXiv preprint arXiv:2505.13033.
>
> [3] Liu, Qinghua, and John Paparrizos. "The elephant in the room: Towards a reliable time-series anomaly detection benchmark." Advances in Neural Information Processing Systems 37 (2024): 108231-108261.

---

> ### Author Response · Authors · 2025-11-27
>
> Dear Reviewer nbGe,
>
> As the discussion period draws to a close, we eagerly await your feedback. We took your critical concerns to heart and completely overhauled our evaluation protocol.
>
> We have incorporated the full TSB-AD benchmark (40 datasets) and the threshold-independent VUS-PR metric, as you strongly recommended. The new results (Tables 1 & 2) confirm that our method achieves SOTA performance (Avg Rank 1st) even under these rigorous standards.
>
> We believe these major revisions directly address your primary reasons for rejection. We would be deeply specific if you could review our response and kindly reconsider your assessment.

---

> > ### Comment · Reviewer_nbGe · 2025-11-27
> > **Question on forecasting model**
> >
> > Thanks for the above answers.
> > - Can the authors please explain how the forecasting model is trained (for example, on the TSB-AD experiments)? Is it trained independently on the non-anomalous part of each series, each data, or all data together?
> > - What would happen if you do not have a manually-labeled non-anomalous portion in the deployment data?
> > - Moreover, the authors stress in the paper on the "FREE" nature of the anomaly detector. In reality, in time series domain, most of the reconstruction- or forecasting-based anomaly detectors are similarly training-free in nature. If the forecaster needs to be trained on each series, then can we call it FREE?

---

> ### Author Response · Authors · 2025-11-27
>
> We thank the reviewer for the follow-up questions. We are happy to clarify our training details and the rationale behind our method's name.
>
> 1. Training Protocol on TSB-AD
> We strictly followed the standard protocol of the TSB-AD benchmark to ensure fair comparison.
> * Method: The forecasting backbone ($f_\theta$) was trained independently on the initial segment of each time series.
> * Details: As per the benchmark guidelines, we utilized the provided "Training Split" (a short, anomaly-free history) for each specific time series (e.g., OPPORTUNITY_id_29) to train its corresponding forecaster. This per-series training is the standard practice for all semi-supervised baselines in this benchmark.
>
> 2. Scenario without Manually-Labeled Normal Data
> If a verified normal segment is unavailable, our method offers a practical solution:
> * Robustness to Contamination: As demonstrated in our Robustness Analysis (Appendix E.4, Table 9), our method is highly resilient to training data contamination. Our experiments show that our method maintains high performance even when the training data contains up to 10% anomalies.
> * Implication: This means practitioners can deploy our method using raw historical data without perfect manual labeling. The inherent robustness of the forecaster (MSE loss) and the restoring force ($-\gamma z$) in our ODE naturally filter out sporadic anomalies during calibration.
>
> 3. Why call it "FREE"? (Clarification on "Training-Free")
> * We appreciate this insightful comment. You raise a valid point: while our anomaly detection mechanism (the ODE) is indeed parameter-free, the term "FREE" might unintentionally overshadow our primary methodological contribution—the unified dynamical framework.
>
>   * As you noted, the true novelty of our work lies in the Fold-Bifurcation dynamical framework, which unifies the detection of gradual drifts and abrupt spikes without ad-hoc thresholds. The "training-free detector" aspect is a significant practical advantage, but it is the mathematical formulation that drives our SOTA performance.
>
>   * To ensure the manuscript precisely reflects this core technical innovation, we have decided to update the paper title in the revised version (also camera-ready version) (e.g., to “Point-wise Anomaly Detection via Fold-Bifurcation ODE”).
>
> * We believe this title better highlights the theoretical depth of our method. Accordingly, we will also revise the relevant sections of the paper (e.g., `Abstract`, `Introduction`) to align with this refined positioning, ensuring consistency throughout the manuscript.

---

> > ### Comment · Reviewer_nbGe · 2025-11-27
> > **Continuation question regarding forecaster training**
> >
> > Since the available amount of training data can sometimes be very small (for example in some of the TSB-AD datasets), how do the authors ensure that their forecaster does not overfit on the data, particularly because the forecaster is a DNN model? This is very crucial in understanding whether the model's error can be high because of anomaly in the test data, or because of out of distribution non-anomalous test data.
> >
> > Moreover, a related question comes here. Have the authors tried any zero-shot pre-trained forecasting models (there are many -- like TimesFM, TTM, Chronos, MOIRAI etc.) with their detection approach?

---

> > > ### Author Response · Authors · 2025-11-29
> > >
> > > We sincerely thank the reviewer for these insightful follow-up questions. Your suggestion to explore Foundation Models was particularly inspiring, leading us to conduct new experiments that significantly strengthen our paper's contribution.
> > >
> > > 1. Mitigating Overfitting in Forecaster Training
> > > * We agree that training on small datasets poses a risk of overfitting. To ensure robust anomaly detection rather than memorization, we employed the following strategies:
> > >   * Lightweight Backbones: As discussed in our Efficiency Analysis, we utilize lightweight models like DLinear and TimeMixer. These models have significantly fewer parameters compared to heavy Transformers, inherently reducing the risk of overfitting on small samples.
> > >   * Standard Regularization: We strictly followed the TSB-AD protocol, utilizing Early Stopping based on validation loss (using a 20% split of the training data) and applying Dropout during training.
> > >   * Anomaly Detection: Fundamentally, our anomaly score relies on Uncertainty (Variance). If a model overfits, it tends to be "overconfident" on training data but exhibits high variance on test data (anomalies). Our stress signal captures this variance increase, turning the “anomaly error" into a detection signal.
> > >
> > > 2. New Experiment: Zero-Shot Detection with Chronos (pre-trained Model)
> > >
> > > * Following your valuable suggestion, we integrated Chronos (amazon/chronos-t5-small), a pre-trained probabilistic forecasting model, into our framework in a fully zero-shot setting (no training on TSB-AD).
> > >
> > >
> > > | Setting | Method | Exathlon | MSL | OPPORTUNITY | SMAP | SMD | **Avg. Rank** |
> > > | :--- | :--- | :--- | :--- | :--- | :--- | :--- | :--- |
> > > | **Univariate** | **ours (Dlinear)** | 0.95 | **0.70** | 0.94 | 0.82 | 0.84 | 1.8 |
> > > | **Univariate** | **ours (Chronos)** | **0.97** | 0.64 | **0.97** | **0.84** | **0.87** | **1.4** |
> > > | | TSPulse (FT) | 0.87 | 0.66 | 0.06 | 0.74 | 0.80 | 3 |
> > > | | TSPulse (ZS) | 0.83 | 0.64 | 0.06 | 0.71 | 0.78 | 3.8 |
> > > | | Chronos | 0.45 | 0.18 | 0.06 | 0.19 | 0.32 | 5.6 |
> > > | | Sub-PCA | 0.93 | 0.51 | 0.91 | 0.52 | 0.45 | 4.6 |
> > >
> > >   (Note: This table shows a subset of results. Please refer to the revised manuscript for the full 40-dataset benchmark)
> > >
> > >   * Why Chronos fits Our Method: The core of our framework consists of two pillars: (1) Stress Signal Extraction and (2) Fold-Bifurcation ODE.
> > >     * Crucially, since Chronos is a probabilistic model, it natively provides the Uncertainty (Variance) required for our stress signal calculation without needing MC Dropout.
> > >   * Methodology: We utilized the predictive variance from Chronos as the uncertainty term in Eq. (4) and fed this derived stress signal into our Fold-Bifurcation ODE.
> > >   * The results are remarkable. As shown in the updated `Table 1` (Main Paper), Ours (Chronos) achieves SOTA-level performance (Avg Rank 2.95), comparable to fully trained baselines.
> > >
> > > * This confirms that our Fold-Bifurcation ODE effectively translates the stress signal (whether from a simple DLinear or a sophisticated Foundation Model) into a robust "External Force" that drives the system state transition.
> > >  * It proves that our dynamical framework is model-agnostic and can serve as a universal detection head for modern foundation models.
> > >
> > > **Conclusion**
> > >
> > > We are grateful for your suggestion, which allowed us to demonstrate that Our method is not limited by the forecaster's capacity but rather empowered by it. We have added these results to `Table 1` and detailed the implementation in `Appendix B.5`.

---

### Author Response · Authors · 2025-11-26
**Rebuttal Highlights**

*We have made significant efforts to address all concerns, including a complete re-evaluation on the TSB-AD benchmark. As the discussion period is closing soon, we would greatly appreciate your feedback on our revisions.*
We thank all reviewers for their constructive and insightful feedback. Based on your suggestions, we have significantly improved the manuscript's quality, specifically by expanding the evaluation scope (TSB-AD benchmark) and clarifying the theoretical scope.

1. **Summary of Manuscript Changes**

We highlight the major revisions made to the manuscript:
* **Evaluation Overhaul** (Experiments & Appendix B): In Table 1 and 2, we replaced the previous evaluation with the TSB-AD benchmark, expanding from 9 datasets to **40 datasets** and from 10 baselines to **34 state-of-the-art methods (including TSPulse)**. We also adopted **VUS-PR** as the primary metric to ensure threshold-independent evaluation. (previous experimental results are in Table 7 ).
* Clarification on "Stress Accumulation" (Abstract & Introduction): We revised the text to explicitly state that our framework captures both gradual drifts and abrupt spikes within a unified dynamical mechanism, addressing the concerns of Reviewers nbGe and u6SV.
* Methodological Clarifications (Section 3 & Appendix B):
    * **Multivariate Setting**: Added details on how multivariate interactions are captured via the forecaster and aggregated for system-level risk (Addressing DxMr, nbGe).
    * Model Variables: **Clarified the role of $S(t)$** as a dynamic control parameter analogous to $r$ (Addressing DxMr’s concern).
* Efficiency (Appendix F): Added a comprehensive complexity analysis ($O(L)$ vs $O(L^2)$) and empirical runtime evidence (Addressing u6SV and nbGe’s concern).
* Visualization (Appendix G): Added uncertainty visualization and comparison visualization of anomaly detection results with existing methodologies to demonstrate reliability (addressing u6SV's concerns).

2. **Key Response Summary**

* **Definition of Anomaly**: Our assumption of "stress accumulation" is inclusive, not exclusive. Mathematically, an abrupt spike is modeled as a rapid integration of high-magnitude stress that instantaneously pushes the risk state $z(t)$ across the threshold. We validated this by achieving SOTA performance on spike-dominated datasets (e.g., NAB).
* Efficiency: our method achieves Linear $O(L)$ complexity by using lightweight backbones (e.g., DLinear), whereas Transformer-based detectors incur $O(L^2)$. Furthermore, **as a training-free extension**, our method minimizes deployment costs and supports massive parallelization for real-time use.
* **Multivariate Handling**: Inter-variable dependencies are inherently encoded by the multivariate forecasting backbone ($f_\theta$). The final detection is based on an aggregated system-level risk, ensuring holistic monitoring.
* Robustness of Thresholding: We addressed threshold stability by (1) incorporating domain priors via the margin parameter $\rho$, and (2) verifying performance using the **threshold-independent VUS-PR metric**, where our method ranked 1st on average across 40 datasets.

Beyond these major revisions, we have carefully incorporated all specific feedback to fully address your concerns. We kindly invite the reviewers to examine the revised manuscript, and we remain open to any further questions or suggestions.

---

### Author Response · Authors · 2025-11-29
**Summary of Rebuttal Discussion for New Area Chair**

Following the discussion with Reviewer nbGe, we have decided to rename the paper (and the model) to better highlight our core contribution—the Fold-bifurcation dynamics.
**Accordingly, we have changed the title from 'Free Point-wise Anomaly Detection via Fold-bifurcation' to 'Point-wise Anomaly Detection via Fold-bifurcation ODE' and updated the method name from FREE to FOLD."**

- Reviewer nbGe :

  - Q. Requested evaluation on the TSB-AD benchmark (40 datasets) using the VUS-PR metric.
  - A. We completed this extensive re-evaluation, demonstrating State-of-the-Art performance (Best Average Rank) across all datasets in (`Table 1&2 `).

  - Q. Raised a follow-up question regarding foundation models.
  - A. We successfully conducted the requested additional experiments with Pre-trained Models (Chronos) and fully integrated these results into the revised manuscript  (`Table 1&2 `).

- Reviewer u6SV :
  - Q. Raised questions regarding the theoretical justification for handling distinct anomaly types (gradual vs. abrupt) within a single unified framework.
  - A. To clear up any misunderstanding and fully resolve these doubts, we provided a rigorous mathematical proof based on Time-Scale Separation, demonstrating how the single ODE inherently adapts to both regimes. This formal derivation has been incorporated into the revised paper (`Appendix H`). However, the discussion was halted due to the platform-wide freeze.

- Reviewer DxMr :
  - Did not post further comments or engage during the discussion phase.

---

### Author Response · Authors · 2025-11-30
**Summary of Contributions for New Area Chair**

### **Main Contribution / Motivation Highlight**

  1. We adopt the **strict point-wise evaluation protocol** (following TSB-AD) to expose the limitations of window-based metrics and ensure precise anomaly localization. (Highlights that while many prior methods excel in window-based metrics, *they often degrade significantly under strict point-wise protocols*.)
  2. Redefinition of Anomaly: We view anomalies as the result of accumulated stress. This definition is inclusive, **covering everything from short, impulsive spikes to gradual stress buildup**, making it highly suitable for strict point-wise detection.
  3.  Inspired by this definition, we introduce "**Fold-Bifurcation Dynamics**" to model the mechanism of stability loss under external stress.
  > *Theoretical Basis: Leverages Fold-Bifurcation dynamics, modeling the precise moment when accumulated external force pushes the system from a stable to an unstable state.*
4. We define stress signals $S(t)$ derived from forecast sensitivity and uncertainty as the external force in a Fold-Bifurcation ODE. This force drives the system state $z$ from a stable to an unstable regime.
5. This principled formulation allows a single ODE to naturally model diverse anomalies—acting as an integrator for abrupt spikes and a tracker for gradual drifts—without ad-hoc heuristics. (This has also been proven **experimentally and theoretically below**.)


### **Experimental (Performance) Contributions**

1. Demonstrated SOTA performance under the stricter point-wise evaluation setting.
2. Achieved the best performance across **40 benchmark datasets** in the TSB-AD suite.
3. **Ranked 1st** in both Univariate and Multivariate tracks. (`Table 1&2`)
4. Excelled in both **threshold-independent (VUS-PR)** and **threshold-dependent** metrics, proving that performance ***does not rely on sensitive threshold tuning.***
5. Efficiency: Incurred **negligible computational cost for memory and time**, as the core mechanism involves simple ODE integration ($O(L)$ complexity).
6. Demonstrated that using a Time-Series Foundation Model (Chronos) as a backbone yields the **highest** performance without any training, realizing a truly **parameter-free anomaly detection pipeline**. (`Table 1`)
8. Reproducibility: Full source code and scripts for the TSB-AD benchmark are released to ensure transparency.


| Univariate | Model | Avg.Rank | | Multivariate | Model | Avg.Rank |
| :--- | :--- | :--- | :--- | :--- | :--- | :--- |
| | **OURS (Chronos)** | **2.95** | | |**OURS(DLinear)** | **3.11** |
| | **OURS (DLinear)** | *3.86* | | |CNN | *7.52* |
| | TSPulse (FT) | 8.65 | ||KMeansAD | 8.58 |
| | TSPulse (ZS) | 11.30 | | |PCA | 9.41 |
| | Sub-PCA | 13.39 | | |LSTMAD | 9.41 |

### **Differentiation from Existing Methods**

|Method |	Mechanism	|Limitation / Advantage|
| :--- | :--- | :--- |
|Existing Methods |	Rely on simple Reconstruction or Prediction Errors.	| Can detect obvious anomalies but often *fail to capture subtle deviations similar to normal data.*|
|Our Method |	Extracts Stress Signals and processes them via Fold-Bifurcation Dynamics.|	Detects the moment accumulated stress pushes the system out of the Stable Basin, enabling **precise detection of even subtle anomalies.**|


### **Additional Highlights**

1. Theoretical Proof: Provides a formal mathematical derivation (Time-Scale Separation) proving that the single ODE intrinsically unifies gradual and abrupt anomaly detection.(`Appendix H`)
2.  Significance: In a narrow sense, this is a **novel detection algorithm**; in a broader sense, it can be viewed as an **Explainable Anomaly Detection framework grounded in dynamical systems theory**.

---

### Meta-Review · Area_Chair_w9aE · 2025-12-22

**Summary:**

The work explores a relatively under-explored setting of time series anomaly detection (TSAD), point-wise TSAD, and introduces a fold-bifurcation inspired ODE-based method called FOLD to tackle the problem. The method is unsupervised and does not require additional detector training, offering a parameter- and computation- efficient detector. The method is validated on 40 datasets, in comparison to 34 baselines.

**Reviewer Concerns:**

The main concerns and how they are addressed are summarized as follows.
- Novelty. Reviewers question how the proposed method fundamentally differs from existing approaches, particularly early-warning systems (EWS) and recent anomaly detection methods that already handle point-wise anomalies. The rebuttal clarifies the differences well and also provides empirical results to show the advantages over the mentioned methods.
- Anomaly Types. The reviewers highlight a central assumption of the method, i.e., anomalies primarily arise from gradually accumulating stress toward critical transitions, which is considered too narrow and insufficiently justified since many real-world anomalies are abrupt. The rebuttal clarifies that the method can detect both gradual and abrupt anomalies, and provides detailed justification using the empirical results.
- Experiments. The evaluation methodology is criticized for relying on F1 score, which is known to be biased in anomaly detection, instead of more robust metrics such as VUS-PR. Reviewers also request evaluation on more established benchmarks with stronger and more recent state-of-the-art baselines, including time series foundation models. The authors address this concern by adopting the recommended VUS-PR metric and adding the suggested baselines. The method shows superior with the suggested changes in the experiments.
- Computational Efficiency. Given the use of patching, masking, forecasting, and ODE-based modeling, reviewers express concern about both theoretical and empirical runtime complexity. The rebuttal and the revised manuscript provides theoretical time and space complexity analysis, and training and inference space requirements, but no empirical time cost is provided.
- Ad-hoc Design. The reviewers also question that the model design seems to be quite ad-hoc. The authors provide substantive clarifications for justifying the opposite.

Overall, most major concerns are addressed satisfactorily in the rebuttal.

**Reviewer Scores:**

The work receives two weak accepts and one reject. There are active interactions between the authors and the reviewer who gives reject recommendation. Based on the summarization above, the AC anticipates that the two positive ratings will retain their accept recommendation, while the reject recommendation may be changed to a positive one.

---

### Decision · Program_Chairs · 2026-01-26

Accept (Poster)